# NEURAL OPTIMAL TRANSPORT

**Alexander Korotin**
Skolkovo Institute of Science and Technology
Artificial Intelligence Research Institute
Moscow, Russia
a.korotin@skoltech.ru

**Daniil Selikhanovych**
Skolkovo Institute of Science and Technology
Moscow, Russia
selikhanovychdaniil@gmail.com

**Evgeny Burnaev**
Skolkovo Institute of Science and Technology
Artificial Intelligence Research Institute
Moscow, Russia
e.burnaev@skoltech.ru

## ABSTRACT

We present a novel neural-networks-based algorithm to compute optimal transport maps and plans for strong and weak transport costs. To justify the usage of neural networks, we prove that they are universal approximators of transport plans between probability distributions. We evaluate the performance of our optimal transport algorithm on toy examples and on the unpaired image-to-image translation.

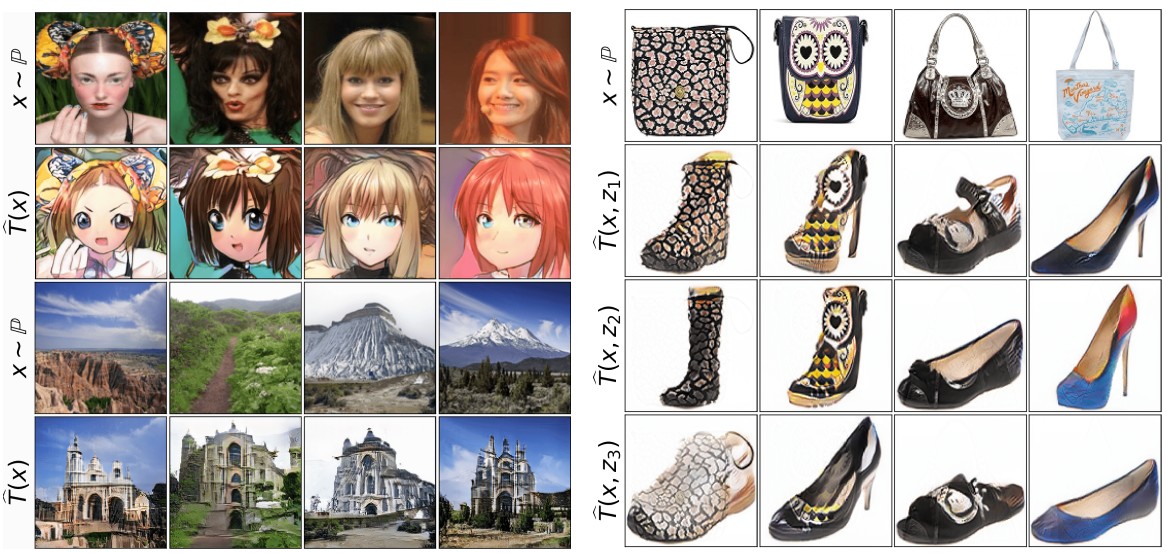

(a) Celeba (female) → anime, outdoor → church, deterministic (one-to-one, $\mathbb{W}_2$).

(b) Handbags → shoes, stochastic (one-to-many, $\mathcal{W}_{2,1}$).

Figure 1: Unpaired translation with our Neural Optimal Transport (NOT) Algorithm 1.

## 1 INTRODUCTION

Solving optimal transport (OT) problems with neural networks has become widespread in machine learning tentatively starting with the introduction of the large-scale OT (Seguy et al., 2017) and Wasserstein GANs (Arjovsky et al., 2017). The majority of existing methods compute the **OT cost** and use it as the loss function to update the generator in generative models (Gulrajani et al., 2017; Liu et al., 2019; Sanjabi et al., 2018; Petzka et al., 2017). Recently, (Rout et al., 2022; Daniels et al., 2021) have demonstrated that the **OT plan** itself can be used as a generative model providing comparable performance in practical tasks.

In this paper, we focus on the methods which compute the OT plan. Most recent methods (Korotin et al., 2021b; Rout et al., 2022) consider OT for the quadratic transport cost (the Wasserstein-2 distance, $\mathbb{W}_2$) and recover a *nonstochastic* OT plan, i.e., a *deterministic* **OT map**. In general, it may

not exist. (Daniels et al., 2021) recover the entropy-regularized stochastic plan, but the procedures for learning the plan and sampling from it are extremely time-consuming due to using score-based models and the Langevin dynamics (Daniels et al., 2021, §6).

**Contributions.** We propose a novel algorithm to compute deterministic and stochastic OT plans with deep neural networks (§4.1, §4.2). Our algorithm is designed for weak and strong optimal transport costs (§2) and generalizes previously known scalable approaches (§3, §4.3). To reinforce the usage of neural nets, we prove that they are universal approximators of transport plans (§4.4). We show that our algorithm can be applied to large-scale computer vision tasks (§5).

**Notations.** We use $\mathcal{X}, \mathcal{Y}, \mathcal{Z}$ to denote Polish spaces and $\mathcal{P}(\mathcal{X}), \mathcal{P}(\mathcal{Y}), \mathcal{P}(\mathcal{Z})$ to denote the respective sets of probability distributions on them. We denote the set of probability distributions on $\mathcal{X} \times \mathcal{Y}$ with marginals $\mathbb{P}$ and $\mathbb{Q}$ by $\Pi(\mathbb{P}, \mathbb{Q})$. For a measurable map $T : \mathcal{X} \times \mathcal{Z} \to \mathcal{Y}$ (or $T : \mathcal{X} \to \mathcal{Y}$), we denote the associated push-forward operator by $T_\#$.

## 2 PRELIMINARIES

In this section, we provide key concepts of the OT theory (Villani, 2008; Santambrogio, 2015; Gozlan et al., 2017; Backhoff-Veraguas et al., 2019) that we use in our paper.

**Strong OT formulation**. For $\mathbb{P} \in \mathcal{P}(\mathcal{X})$, $\mathbb{Q} \in \mathcal{P}(\mathcal{Y})$ and a cost function $c : \mathcal{X} \times \mathcal{Y} \to \mathbb{R}$, Monge's primal formulation of the OT cost is

$$\text{Cost}(\mathbb{P}, \mathbb{Q}) \stackrel{\text{def}}{=} \inf_{T_\#\mathbb{P}=\mathbb{Q}} \int_{\mathcal{X}} c(x, T(x)) d\mathbb{P}(x), \quad (1)$$

where the minimum is taken over measurable functions (transport maps) $T : \mathcal{X} \to \mathcal{Y}$ that map $\mathbb{P}$ to $\mathbb{Q}$ (Figure 2). The optimal $T^*$ is called the OT *map*.

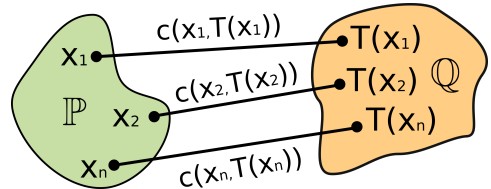

Figure 2: Monge's OT formulation.

Note that (1) is not symmetric and does not allow mass splitting, i.e., for some $\mathbb{P}, \mathbb{Q} \in \mathcal{P}(\mathcal{X}), \mathcal{P}(\mathcal{Y})$, there may be no $T$ satisfying $T_\#\mathbb{P} = \mathbb{Q}$. Thus, (Kantorovitch, 1958) proposed the following relaxation:

$$\text{Cost}(\mathbb{P}, \mathbb{Q}) \stackrel{\text{def}}{=} \inf_{\pi \in \Pi(\mathbb{P}, \mathbb{Q})} \int_{\mathcal{X} \times \mathcal{Y}} c(x, y) d\pi(x, y), \quad (2)$$

where the minimum is taken over all transport plans $\pi$ (Figure 3a), i.e., distributions on $\mathcal{X} \times \mathcal{Y}$ whose marginals are $\mathbb{P}$ and $\mathbb{Q}$. The optimal $\pi^* \in \Pi(\mathbb{P}, \mathbb{Q})$ is called the optimal transport *plan*. If $\pi^*$ is of the form $[\text{id}, T^*]_\#\mathbb{P} \in \Pi(\mathbb{P}, \mathbb{Q})$ for some $T^*$, then $T^*$ minimizes (1). In this case, the plan is called *deterministic*. Otherwise, it is called *stochastic* (nondeterministic).

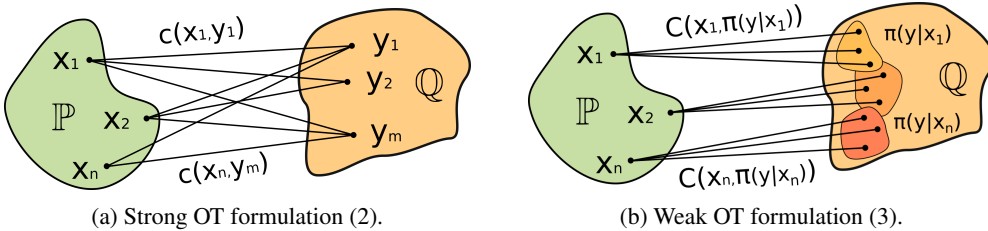

(a) Strong OT formulation (2).  (b) Weak OT formulation (3).

Figure 3: Strong (Kantorovich's) and weak (Gozlan et al., 2017) optimal transport fomulations.

An example of OT cost for $\mathcal{X} = \mathcal{Y} = \mathbb{R}^D$ is the ($p$-th power of) Wasserstein-$p$ distance $\mathbb{W}_p$, i.e., formulation (2) with $c(x, y) = \|x - y\|^p$. Two its most popular cases are $p = 1, 2$ ($\mathbb{W}_1, \mathbb{W}_2^2$).

**Weak OT formulation** (Gozlan et al., 2017). Let $C : \mathcal{X} \times \mathcal{P}(\mathcal{Y}) \to \mathbb{R}$ be a *weak* cost, i.e., a function which takes a point $x \in \mathcal{X}$ and a distribution of $y \in \mathcal{Y}$ as input. The weak OT cost between $\mathbb{P}, \mathbb{Q}$ is

$$\text{Cost}(\mathbb{P}, \mathbb{Q}) \stackrel{\text{def}}{=} \inf_{\pi \in \Pi(\mathbb{P}, \mathbb{Q})} \int_{\mathcal{X}} C(x, \pi(\cdot|x)) d\pi(x), \quad (3)$$

where $\pi(\cdot|x)$ denotes the conditional distribution (Figure 3b). Note that (3) is a generalization of (2). Indeed, for cost $C(x, \mu) = \int_{\mathcal{Y}} c(x, y) d\mu(y)$, the weak formulation (3) becomes strong (2). An

example of a weak OT cost for $\mathcal{X} = \mathcal{Y} = \mathbb{R}^D$ is the $\gamma$-weak ($\gamma \geq 0$) Wasserstein-2 ($\mathcal{W}_{2,\gamma}$):

$$C(x, \mu) = \int_{\mathcal{Y}} \frac{1}{2}\|x - y\|^2 d\mu(y) - \frac{\gamma}{2}\text{Var}(\mu) \tag{4}$$

**Existence and duality.** Throughout the paper, we consider weak costs $C(x, \mu)$ which are lower bounded, convex in $\mu$ and jointly lower semicontinuous in an appropriate sense. Under these assumptions, (Backhoff-Veraguas et al., 2019) prove that the minimizer $\pi^*$ of (3) always exists.[1] With mild assumptions on $c$, strong costs satisfy these assumptions. In particular, they are linear w.r.t. $\mu$, and, consequently, convex. The $\gamma$-weak quadratic cost (4) is lower-bounded (for $\gamma \leq 1$) and is also convex since the functional $\text{Var}(\mu)$ is concave in $\mu$. For the costs in view, the *dual form* of (3) is

$$\text{Cost}(\mathbb{P}, \mathbb{Q}) = \sup_f \int_{\mathcal{X}} f^C(x) d\mathbb{P}(x) + \int_{\mathcal{Y}} f(y) d\mathbb{Q}(y), \tag{5}$$

where $f$ are the upper-bounded continuous functions with not very rapid growth (Backhoff-Veraguas et al., 2019, Equation 1.2) and $f^C$ is the weak $C$-transform of $f$, i.e.

$$f^C(x) \stackrel{\text{def}}{=} \inf_{\mu \in \mathcal{P}(\mathcal{Y})} \left\{ C(x, \mu) - \int_{\mathcal{Y}} f(y) d\mu(y) \right\}. \tag{6}$$

Note that for strong costs $C$, the infimum is attained at any $\mu \in \mathcal{P}(\mathcal{Y})$ supported on the $\arg\inf_{y \in \mathcal{Y}}\{c(x, y) - f(y)\}$ set. Therefore, it suffices to use the strong $c$-transform:

$$f^C(x) = f^c(x) \stackrel{\text{def}}{=} \inf_{y \in \mathcal{Y}} \left\{ c(x, y) - f(y) \right\}. \tag{7}$$

For strong costs (2), formula (5) with (7) is the well known Kantorovich duality (Villani, 2008, §5).

**Nonuniqueness.** In general, an OT plan $\pi^*$ is not unique, e.g., see (Peyré et al., 2019, Remark 2.3).

## 3 RELATED WORK

In large-scale machine learning, **OT costs** are primarily used as the loss to learn generative models. Wasserstein GANs introduced by (Arjovsky et al., 2017; Gulrajani et al., 2017) are the most popular examples of this approach. We refer to (Korotin et al., 2022b; 2021b) for recent surveys of principles of WGANs. However, these models are *out of scope* of our paper since they only compute the OT cost but not OT plans or maps (§4.3). To compute **OT plans** (or maps) is a more challenging problem, and only a limited number of scalable methods to solve it have been developed.

We overview methods to compute OT plans (or maps) below. We emphasize that existing methods are designed only for *strong* OT formulation (2). Most of them search for a deterministic solution (1), i.e., for a map $T^*$ rather than a stochastic plan $\pi^*$, although $T^*$ might not always exist.

To compute the OT plan (map), (Lu et al., 2020; Xie et al., 2019) approach the **primal** formulation (1) or (2). Their methods imply using generative models and yield complex optimization objectives with several adversarial regularizers, e.g., they are used to enforce the boundary condition ($T_\#\mathbb{P} = \mathbb{Q}$). As a result, the methods are hard to setup since they require careful selection of hyperparameters.

In contrast, methods based on the **dual** formulation (5) have simpler optimization procedures. Most of such methods are designed for OT with the quadratic cost, i.e., the Wasserstein-2 distance ($\mathbb{W}_2^2$). An evaluation of these methods is provided in (Korotin et al., 2021b). Below we mention their issues.

Methods by (Taghvaei & Jalali, 2019; Makkuva et al., 2020; Korotin et al., 2021a;c) based on input-convex neural networks (ICNNs, see (Amos et al., 2017)) have solid theoretical justification, but do not provide sufficient performance in practical large-scale problems. Methods based on entropy regularized OT (Genevay et al., 2016; Seguy et al., 2017; Daniels et al., 2021) recover regularized OT plan that is *biased* from the true one, it is hard to sample from it or compute its density.

According to (Korotin et al., 2021b), the best performing approach is $\lceil$MM:R$\rfloor$, which is based on the maximin reformulation of (5). It recovers OT maps fairly well and has a good generative performance. The follow-up papers (Rout et al., 2022; Fan et al., 2022) test extensions of this approach for more general strong transport costs $c(\cdot, \cdot)$ and apply it to compute $\mathbb{W}_2$ barycenters (Korotin et al., 2022a). Their key limitation is that it aims to recover a *deterministic* OT map $T^*$ which might not exist.

---

[1] Backhoff-Veraguas et al. (2019) work with the subset $\mathcal{P}_p(\mathcal{Y}) \subset \mathcal{P}(\mathcal{Y})$ whose $p$-th moment is finite. Henceforth, we also work in $\mathcal{P}_p(\mathcal{Y})$ equipped with the Wasserstein-$p$ topology. Since this detail is not principal for our subsequent analysis, to keep the exposition simple, we still write $\mathcal{P}(\mathcal{Y})$ but actually mean $\mathcal{P}_p(\mathcal{Y})$.

## 4    ALGORITHM FOR LEARNING OT PLANS

In this section, we develop a novel neural algorithm to recover a solution $\pi^*$ of OT problem (3). The following lemma will play an important role in our derivations.

**Lemma 1** (Existence of transport maps.). *Let $\mu$ and $\nu$ be probability distributions on $\mathbb{R}^M$ and $\mathbb{R}^N$. Assume that $\mu$ is atomless. Then there exists a measurable $t : \mathbb{R}^M \to \mathbb{R}^N$ satisfying $t_\# \mu = \nu$.*

*Proof.* (Santambrogio, 2015, Cor. 1.29) proves the fact for $M = N$. The proof works for $M \neq N$. $\square$

Throughout the paper we assume that $\mathbb{P}, \mathbb{Q}$ are supported on subsets $\mathcal{X} \subset \mathbb{R}^P$, $\mathcal{Y} \subset \mathbb{R}^Q$, respectively.

### 4.1    REFORMULATION OF THE DUAL PROBLEM

First, we reformulate the optimization in $C$-transform (6). For this, we introduce a subset $\mathcal{Z} \subset \mathbb{R}^S$ with an atomless distribution $\mathbb{S}$ on it, e.g., $\mathbb{S} = \text{Uniform}([0, 1])$ or $\mathcal{N}(0, 1)$.

**Lemma 2** (Reformulation of the $C$-transform). *The following equality holds:*

$$f^C(x) = \inf_t \left\{ C(x, t_\# \mathbb{S}) - \int_{\mathcal{Z}} f(t(z)) d\mathbb{S}(z) \right\}, \tag{8}$$

*where the infimum is taken over all measurable $t : \mathcal{Z} \to \mathcal{Y}$.*

*Proof.* For all $x \in \mathcal{X}$ and $t : \mathcal{Z} \to \mathcal{Y}$ we have $f^C(x) \leq C(x, t_\# \mathbb{S}) - \int_{\mathcal{Z}} f(t(z)) d\mathbb{S}(z)$. The inequality is straightforward: we substitute $\mu = t_\# \mathbb{S}$ to (6) to upper bound $f^C(x)$ and use the change of variables. Taking the infimum over $t$, we obtain

$$f^C(x) \leq \inf_t \left\{ C(x, t_\# \mathbb{S}) - \int_{\mathcal{Z}} f(t(z)) d\mathbb{S}(z) \right\}. \tag{9}$$

Now let us turn (9) to an equality. We need to show that $\forall \epsilon > 0$ there exists $t^\epsilon : \mathcal{Z} \to \mathcal{Y}$ satisfying

$$f^C(x) + \epsilon \geq C(x, t^\epsilon_\# \mathbb{S}) - \int_{\mathcal{Z}} f(t^\epsilon(z)) d\mathbb{S}(z). \tag{10}$$

By (6) and the definition of inf, $\exists \mu^\epsilon \in \mathcal{P}(\mathcal{Y})$ such that $f^C(x) + \epsilon \geq C(x, \mu^\epsilon) - \int_{\mathcal{Y}} f(y) d\mu^\epsilon(y)$. Thanks to Lemma 1, there exists $t^\epsilon : \mathcal{Z} \to \mathcal{Y}$ such that $\mu^\epsilon = t^\epsilon_\# \mathbb{S}$, i.e., (10) holds true. $\square$

Now we use Lemma 2 to get an analogous reformulation of the integral of $f^C$ in the dual form (5).

**Lemma 3** (Reformulation of the integrated $C$-transform). *The following equality holds:*

$$\int_{\mathcal{X}} f^C(x) d\mathbb{P}(x) = \inf_T \int_{\mathcal{X}} \left( C(x, T(x, \cdot)_\# \mathbb{S}) - \int_{\mathcal{Z}} f(T(x, z)) d\mathbb{S}(z) \right) d\mathbb{P}(x), \tag{11}$$

*where the inner minimization is performed over all measurable functions $T : \mathcal{X} \times \mathcal{Z} \to \mathcal{Y}$.*

*Proof.* The lemma follows from the interchange between the infimum and integral provided by the Rockafellar's interchange theorem (Rockafellar, 1976, Theorem 3A).

The theorem states that for a function $F : \mathcal{A} \times \mathcal{B} \to \mathbb{R}$ and a distribution $\nu$ on $\mathcal{A}$,

$$\int_{\mathcal{A}} \inf_{b \in \mathcal{B}} F(a, b) d\nu(a) = \inf_{H : \mathcal{A} \to \mathcal{B}} \int_{\mathcal{A}} F(a, H(a)) d\nu(a) \tag{12}$$

We apply (12), use $\mathcal{A} = \mathcal{X}$, $\nu = \mathbb{P}$, and put $\mathcal{B}$ to be the space of measurable functions $\mathcal{Z} \to \mathcal{Y}$, and $F(a, b) = C(a, b_\# \mathbb{S}) - \int_{\mathcal{Y}} f(y) d[b_\# \mathbb{S}](y)$. Consequently, we obtain that $\int_{\mathcal{X}} f^C(x) d\mathbb{P}(x)$ equals

$$\inf_H \int_{\mathcal{X}} \left( C(x, H(x)_\# \mathbb{S}) - \int_{\mathcal{Y}} f(y) d[H(x)_\# \mathbb{S}](y) \right) d\mathbb{P}(x) \tag{13}$$

Finally, we note that the optimization over functions $H : \mathcal{X} \to \{t : \mathcal{Z} \to \mathcal{Y}\}$ equals the optimization over functions $T : \mathcal{X} \times \mathcal{Z} \to \mathcal{Y}$. We put $T(x, z) = [H(x)](z)$, use the change of variables for $y = T(x, z)$ and derive (11) from (13). $\square$

Lemma 3 provides the way to represent the dual form (5) as a saddle point optimization problem.

**Corollary 1** (Maximin reformulation of the dual problem). *The following holds:*

$$\text{Cost}(\mathbb{P}, \mathbb{Q}) = \sup_f \inf_T \mathcal{L}(f, T), \tag{14}$$

*where the functional $\mathcal{L}$ is defined by*

$$\mathcal{L}(f, T) \stackrel{def}{=} \int_{\mathcal{Y}} f(y) d\mathbb{Q}(y) + \int_{\mathcal{X}} \left( C\big(x, T(x, \cdot)_{\#}\mathbb{S}\big) - \int_{\mathcal{Z}} f\big(T(x, z)\big) d\mathbb{S}(z) \right) d\mathbb{P}(x). \tag{15}$$

*Proof.* It suffices to substitute (11) into (5). □

We say that functions $T : \mathcal{X} \times \mathcal{Z} \to \mathcal{Y}$ are *stochastic maps*. If a map $T$ is independent of $z$, i.e., for all $(x, z) \in \mathcal{X} \times \mathcal{Z}$ we have $T(x, z) \equiv T(x)$, we say the map is *deterministic*.

The idea behind the introduced notation is the following. An optimal transport plan $\pi^*$ might be nondeterministic, i.e., there might not exist a deterministic function $T : \mathcal{X} \to \mathcal{Y}$ which satisfies $\pi^* = [\text{id}_{\mathcal{X}}, T]_{\#}\mathbb{P}$. However, each transport plan $\pi \in \Pi(\mathbb{P}, \mathbb{Q})$ can be represented implicitly through a stochastic function $T : \mathcal{X} \times \mathcal{Z} \to \mathcal{Y}$. This fact is known as **noise outsourcing** (Kallenberg, 1997, Theorem 5.10) for $\mathcal{Z} = [0, 1] \subset \mathbb{R}^1$ and $\mathbb{S} = \text{Uniform}([0, 1])$. Combined with Lemma 1, the noise outsourcing also holds for a general $\mathcal{Z} \subset \mathbb{R}^S$ and atomless $\mathbb{S} \in \mathcal{P}(\mathcal{Z})$. We visualize the idea in Figure 4. For a plan $\pi$, there might exist multiple maps $T$ which represent it.

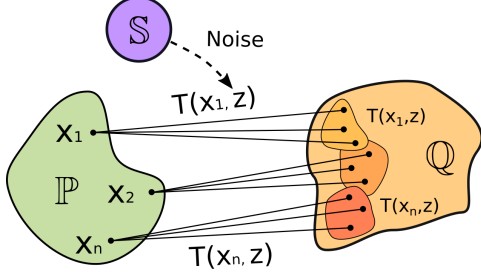

Figure 4: Stochastic function $T(x, z)$ representing a transport plan. The function's input is $x \in \mathcal{X}$ and $z \sim \mathbb{S}$.

For a pair of probability distributions $\mathbb{P}, \mathbb{Q}$, we say that $T^*$ is a *stochastic optimal transport map* if it realizes some optimal transport plan $\pi^*$. Such maps solve the inner problem in (14) for optimal $f^*$.

**Lemma 4** (Optimal maps solve the maximin problem). *For any maximizer $f^*$ of (5) and for any stochastic map $T^*$ which realizes some optimal transport plan $\pi^*$, it holds that*

$$T^* \in \arg\inf_T \mathcal{L}(f^*, T). \tag{16}$$

*Proof.* Let $\pi^*$ be the OT plan realized by $T^*$. We derive

$$\int_{\mathcal{X}} \int_{\mathcal{Z}} f^*\big(T^*(x, z)\big) d\mathbb{S}(z) d\mathbb{P}(x) = \int_{\mathcal{X}} \int_{\mathcal{Y}} f^*(y) d\pi^*(y|x) d\pi^*(x) =$$

$$\int_{\mathcal{X} \times \mathcal{Y}} f^*(y) d\pi^*(x, y) = \int_{\mathcal{Y}} f^*(y) d\mathbb{Q}(y), \tag{17}$$

where we change the variables for $y = T^*(x, z)$ and use the property $d\pi^*(x) = d\mathbb{P}(x)$. Now assume that $T^* \notin \arg\inf_T \mathcal{L}(f^*, T)$. In this case, from the definition (15) we conclude that $\mathcal{L}(f^*, T^*) > \text{Cost}(\mathbb{P}, \mathbb{Q})$. However, we derive substituting (17) into (15), we see that $\mathcal{L}(f^*, T^*) = \int_{\mathcal{X}} C\big(x, \underbrace{T^*(x, \cdot)_{\#}\mathbb{S}}_{\pi^*(y|x)}\big) \underbrace{d\mathbb{P}(x)}_{d\pi^*(x)} = \text{Cost}(\mathbb{P}, \mathbb{Q})$, which is a contradiction. Thus, (16) holds true. □

For the $\gamma$-weak quadratic cost (4) which we use in the experiments (§5), a maximizer $f^*$ of (5) indeed exists, see (Alibert et al., 2019, §5.22) or (Gozlan & Juillet, 2020). Thanks to our Lemma 4, one may solve the saddle point problem (14) and extract an optimal stochastic transport map $T^*$ from its solution $(f^*, T^*)$. In general, the $\arg\inf$ set for $f^*$ may contain not only the optimal stochastic transport maps but other stochastic functions as well. In Appendix F, we show that for strictly convex (in $\mu$) costs $C(x, \mu)$, all the solutions of (14) provide stochastic OT maps.

## 4.2 PRACTICAL OPTIMIZATION PROCEDURE

To approach the problem (14) in practice, we use neural networks $T_\theta : \mathbb{R}^P \times \mathbb{R}^S \to \mathbb{R}^Q$ and $f_\omega : \mathbb{R}^Q \to \mathbb{R}$ to parameterize $T$ and $f$, respectively. We train their parameters with the stochastic gradient ascent-descent (SGAD) by using random batches from $\mathbb{P}, \mathbb{Q}, \mathbb{S}$, see Algorithm 1.

---

**Algorithm 1:** Neural optimal transport (NOT)

---

**Input** : distributions $\mathbb{P}, \mathbb{Q}, \mathbb{S}$ accessible by samples; mapping network $T_\theta : \mathbb{R}^P \times \mathbb{R}^S \to \mathbb{R}^Q$;
potential network $f_\omega : \mathbb{R}^Q \to \mathbb{R}$; number of inner iterations $K_T$;
(weak) cost $C : \mathcal{X} \times \mathcal{P}(\mathcal{Y}) \to \mathbb{R}$; empirical estimator $\widehat{C}(x, T(x, Z))$ for the cost;

**Output :** learned stochastic OT map $T_\theta$ representing an OT plan between distributions $\mathbb{P}, \mathbb{Q}$;

**repeat**

    Sample batches $Y \sim \mathbb{Q}$, $X \sim \mathbb{P}$; for each $x \in X$ sample batch $Z_x \sim \mathbb{S}$;

    $\mathcal{L}_f \leftarrow \frac{1}{|X|} \sum_{x \in X} \frac{1}{|Z_x|} \sum_{z \in Z_x} f_\omega(T_\theta(x, z)) - \frac{1}{|Y|} \sum_{y \in Y} f_\omega(y)$;

    Update $\omega$ by using $\frac{\partial \mathcal{L}_f}{\partial \omega}$;

    **for** $k_T = 1, 2, \ldots, K_T$ **do**

        Sample batch $X \sim \mathbb{P}$; for each $x \in X$ sample batch $Z_x \sim \mathbb{S}$;

        $\mathcal{L}_T \leftarrow \frac{1}{|X|} \sum_{x \in X} \left[ \widehat{C}(x, T_\theta(x, Z_x)) - \frac{1}{|Z_x|} \sum_{z \in Z_x} f_\omega(T_\theta(x, z)) \right]$;

        Update $\theta$ by using $\frac{\partial \mathcal{L}_T}{\partial \theta}$;

**until** *not converged*;

---

Our Algorithm 1 requires an empirical estimator $\widehat{C}$ for $C(x, T(x, \cdot)_{\#} \mathbb{S})$. If the cost is strong, it is straightforward to use the following *unbiased* Monte-Carlo estimator from a random batch $Z \sim \mathbb{S}$:

$$C(x, T(x, \cdot)_{\#} \mathbb{S}) = \int_{\mathcal{Z}} c(x, T(x, z)) d\mathbb{S}(z) \approx \frac{1}{|Z|} \sum_{z \in Z} c(x, T(x, z)) \stackrel{def}{=} \widehat{C}(x, T(x, Z)). \quad (18)$$

For general costs $C$, providing an estimator might be nontrivial. For the $\gamma$-weak quadratic cost (4), such an *unbiased* Monte-Carlo estimator is straightforward to derive:

$$\widehat{C}(x, T(x, Z)) \stackrel{def}{=} \frac{1}{2|Z|} \sum_{z \in Z} \|x - T(x, z)\|^2 - \frac{\gamma}{2} \hat{\sigma}^2, \quad (19)$$

where $\hat{\sigma}^2$ is the (corrected) batch variance $\hat{\sigma}^2 = \frac{1}{|Z|-1} \sum_{z \in Z} \|T(x, z) - \frac{1}{|Z|} \sum_{z \in Z} T(x, z)\|^2$. To estimate strong costs (18), it is enough to sample a single noise vector ($|Z| = 1$). To estimate the $\gamma$-weak quadratic cost (19), one needs $|Z| \geq 2$ since the estimation of the variance $\hat{\sigma}^2$ is needed.

### 4.3 RELATION TO PRIOR WORKS

**Generative adversarial learning**. Our algorithm 1 is a novel approach to learn stochastic OT plans; it is not a GAN or WGAN-based solution endowed with additional losses such as the OT cost. WGANs (Arjovsky et al., 2017) do not learn an OT plan but use the (strong) OT cost as the loss to learn the generator network. Their problem is $\inf_T \sup_f \mathcal{V}(T, f)$. The generator $T^*$ solves the *outer* $\inf_T$ problem and is the *first* coordinate of an optimal saddle point $(T^*, f^*)$. In our algorithm 1, problem (15) is $\sup_f \inf_T \mathcal{L}(f, T)$, the generator (transport map) $T^*$ solves of the *inner* $\inf_T$ problem and is the *second* coordinate of an optimal saddle point $(f^*, T^*)$. Intuitively, in our case the generator $T$ is adversarial to potential $f$ (discriminator), not vise-versa as in GANs. Theoretically, the problem is also *significantly* different – swapping $\inf_T$ and $\sup_f$, in general, yields a different problem with different solutions, e.g., $1 = \inf_x \sup_y \sin(x+y) \neq \sup_y \inf_x \sin(x+y) = -1$. Practically, we do $K_T > 1$ updates of $T$ per one step of $f$, which again differs from common GAN practices, where multiple updates of $f$ are done per a step of $T$. Finally, in contrast to WGANs, we do not need to enforce any constraints on $f$, e.g., the 1-Lipschitz continuity.

**Stochastic generator parameterization.** We add an additional noise input $z$ to transport map $T(x, z)$ to make it stochastic. This approach is a common technical instrument to parameterize one-to-many mappings in generative modeling, see (Almahairi et al., 2018, §3.1) or (Zhu et al., 2017b, §3). In the context of OT, (Yang & Uhler, 2019) employ a stochastic generator to learn a transport plan $\pi$ in the unbalanced OT problem (Chizat, 2017). Due to this, their optimization objective slightly resembles ours (15). However, this similarity is deceptive, see Appendix G.

**Dual OT solvers.** Our algorithm 1 recovers stochastic plans for weak costs (3). It subsumes previously known approaches which learn deterministic OT maps for strong costs (2). When the cost is strong (3) and transport map $T$ is restricted to be deterministic $T(x, z) \equiv T(x)$, our Algorithm 1 yields maximin method $\lceil$MM:R$\rfloor$, which was discussed in (Korotin et al., 2021b, §2) for the quadratic cost $\frac{1}{2}\|x - y\|^2$ and further developed by (Rout et al., 2022) for the $Q$-embedded cost $-\langle Q(x), y \rangle$ and by (Fan et al., 2022) for other strong costs $c(x, y)$. These works are the most related to our study.

### 4.4 Universal Approximation with Neural Networks

In this section, we show that it is possible to approximate transport maps with neural nets.

**Theorem 1** (Neural networks are universal approximators of stochastic transport maps). *Assume that $\mathcal{X}, \mathcal{Z}$ are compact and $\mathbb{Q}$ has finite second moment. Let $T$ be a stochastic map from $\mathbb{P}$ to $\mathbb{Q}$ (not necessarily optimal). Then for any nonaffine continuous activation function which is continuously differentiable at at least one point (with nonzero derivative at that point) and for any $\epsilon > 0$, there exists a neural network $T_\theta : \mathbb{R}^P \times \mathbb{R}^S \to \mathbb{R}^Q$ satisfying*

$$\|T_\theta - T\|_{L^2}^2 \leq \epsilon \qquad and \qquad \mathbb{W}_2^2\big((T_\theta)_\#(\mathbb{P} \times \mathbb{S}), \mathbb{Q}\big) \leq \epsilon, \tag{20}$$

*where $L^2 = L^2(\mathbb{P} \times \mathbb{S}, \mathcal{X} \times \mathcal{Z} \to \mathbb{R}^Q)$ is the space of quadratically integrable w.r.t. $\mathbb{P} \times \mathbb{S}$ functions $\mathcal{X} \times \mathcal{Z} \to \mathbb{R}^Q$. That is, the network $T_\theta$ generates a distribution which is $\epsilon$-close to $\mathbb{Q}$ in $\mathbb{W}_2^2$.*

*Proof.* The squared norm $\|T\|_{L^2}^2$ is equal to the second moment of $\mathbb{Q}$ since $T$ pushes $\mathbb{P} \times \mathbb{S}$ to $\mathbb{Q}$. The distribution $\mathbb{Q}$ has finite second moment, and, consequently, $T \in L^2$. Thanks to (Folland, 1999, Proposition 7.9), the continuous functions $C^0(\mathcal{X} \times \mathcal{Z} \to \mathbb{R}^Q)$ are dense[2] in $L^2$. According to (Kidger & Lyons, 2020, Theorem 3.2), the neural networks $\mathbb{R}^P \times \mathbb{R}^S \to \mathbb{R}^Q$ with the above-mentioned activations are dense in $C^0(\mathcal{X} \times \mathcal{Z} \to \mathbb{R}^Q)$ w.r.t. $L^\infty$ norm and, consequently, w.r.t. $L^2$ norm. Combining these results yields that neural nets are dense in $L^2$, and for every $\epsilon > 0$ there necessarily exists network $T_\theta$ satisfying the left inequality in (20). For $T_\theta$, the right inequality follows from (Korotin et al., 2021a, Lemma A.2). $\qquad\square$

Our Theorem 1 states that neural nets can approximate stochastic maps in $L^2$ norm. It should be taken into account that such continuous nets $T_\theta$ may be highly irregular and hard to learn in practice.

## 5 Evaluation

We perform *comparison* with the weak discrete OT (considered as the ground truth) on toy 2D, 1D distributions in Appendices B, C, respectively. In this section, we test our algorithm on an unpaired image-to-image translation task. We perform *comparison* with popular existing translation methods in Appendix D. The code is written in `PyTorch` framework and is publicly available at

> https://github.com/iamalexkorotin/NeuralOptimalTransport

**Image datasets.** We use the following publicly available datasets as $\mathbb{P}, \mathbb{Q}$: aligned anime faces[3], celebrity faces (Liu et al., 2015), shoes (Yu & Grauman, 2014), Amazon handbags, churches from LSUN dataset (Yu et al., 2015), outdoor images from the MIT places database (Zhou et al., 2014). The size of datasets varies from 50K to 500K images.

**Train-test split.** We pick 90% of each dataset for unpaired training. The rest 10% are considered as the test set. All the results presented here are *exclusively* for test images, i.e., *unseen data*.

**Transport costs.** We experiment with the strong ($\gamma = 0$) and $\gamma$-weak ($\gamma > 0$) quadratic costs. Testing other costs, e.g., perceptual (Johnson et al., 2016) or semantic (Cherian & Sullivan, 2019), might be interesting practically, but these two quadratic costs already provide promising performance.

The other **training details** are given in Appendix E.

### 5.1 Preliminary Evaluation

In the preliminary experiments with *strong* cost ($\gamma = 0$), we noted that $T(x, z)$ becomes independent of $z$. For a fixed potential $f$ and a point $x$, the map $T(x, \cdot)$ learns to be the map pushing distribution $\mathbb{S}$ to some $\arg\inf$ distribution $\mu$ of (6). For strong costs, there are suitable degenerate distributions $\mu$, see the discussion around (7). Thus, for $T$ it becomes unnecessary to keep any dependence on $z$; it simply learns a deterministic map $T(x, z) = T(x)$. We call this behavior a **conditional collapse**.

Importantly, for the $\gamma$-*weak* cost ($\gamma > 0$), we noted a different behavior: the stochastic map $T(x, z)$ did not collapse conditionally. To explain this, we substitute (4) into (3) to obtain

$$\mathcal{W}_{2,\gamma}^2(\mathbb{P}, \mathbb{Q}) = \inf_{\pi \in \Pi(\mathbb{P}, \mathbb{Q})} \left[ \int_{\mathcal{X} \times \mathcal{Y}} \frac{1}{2}\|x - y\|^2 d\pi(x, y) - \gamma \cdot \int_{\mathcal{X}} \frac{1}{2}\text{Var}\big(\pi(y|x)\big) \underbrace{d\pi(x)}_{d\mathbb{P}(x)} \right].$$

The first term is analogous to the strong cost ($\mathbb{W}_2 = \mathcal{W}_{2,0}$), while the additional second term stimulates the OT plan to be stochastic, i.e., to have high conditional variance.

---

[2]The proposition considers scalar-valued functions ($Q = 1$), but is analogous for vector-valued functions.

[3]kaggle.com/reitanaka/alignedanimefaces

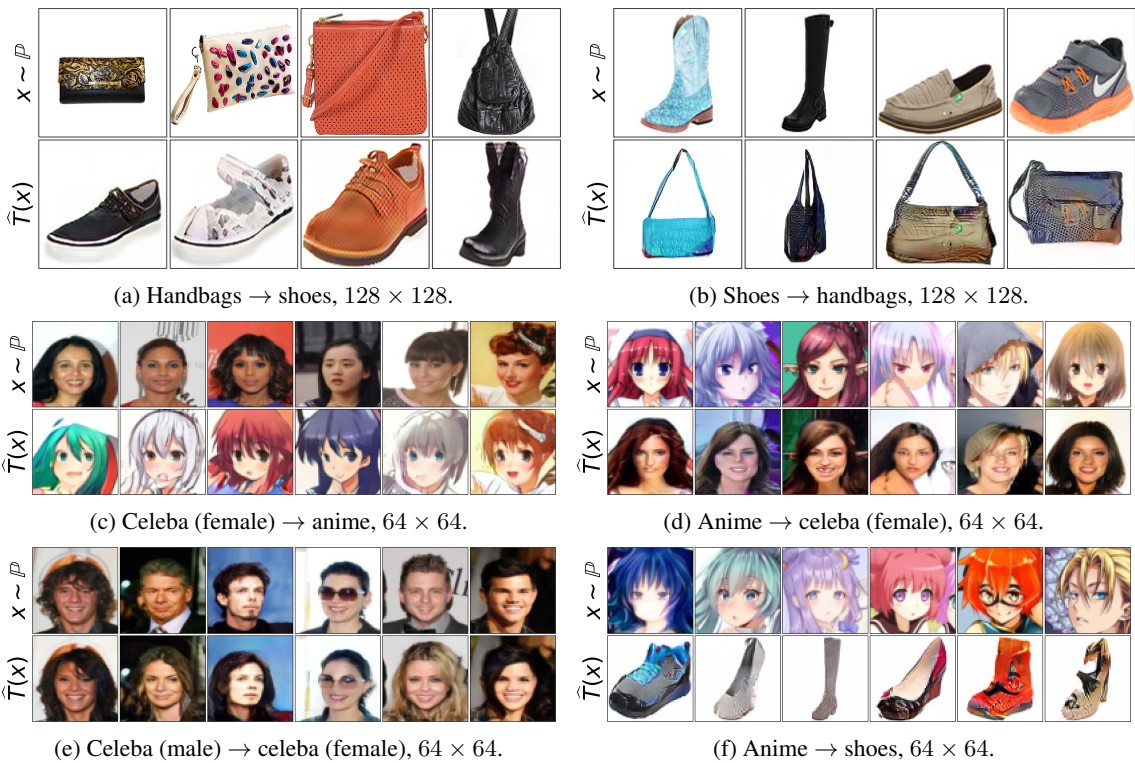

Figure 5: Unpaired translation with deterministic OT maps ($\mathbb{W}_2$).

Taking into account our preliminary findings, we perform two types of experiments. In §5.2, we learn deterministic (one-to-one) translation maps $T(x)$ for the strong cost ($\gamma = 0$), i.e., do not add $z$-channel. In §5.3, we learn stochastic (one-to-many) maps $T(x, z)$ for the $\gamma$-weak cost ($\gamma > 0$). For completeness, in Appendix A, we study how varying $\gamma$ affects the *diversity* of samples.

## 5.2 ONE-TO-ONE TRANSLATION WITH OPTIMAL MAPS

We learn deterministic OT maps between various pairs of datasets. We provide the results in Figures 1a and 5. Extra results for all the dataset pairs that we consider are given in Appendix H.

Being optimal, our translation map $\widehat{T}(x)$ tries to minimally change the image content $x$ in the $L^2$ pixel space. This results in preserving certain features during translation. In *shoes ↔ handbags* (Figures 5b, 5a), the image color and texture of the pushforward samples reflects those of input samples. In *celeba (female) ↔ anime* (Figures 1a, 5c, 5d), head forms, hairstyles are mostly similar for input and output images. The hair in anime is usually bigger than that in celeba. Thus, when translating *celeba (female) ↔ anime*, the anime hair inherits the color from the celebrity image background. In *outdoor → churches* (Figure 1a), the ground and the sky are preserved, in *celeba (male) → celeba (female)* (Figure 5e) – the face does not change. We also provide results for translation in the case when the input and output domains are significantly different, see *anime → shoes* (Figure 5f).

**Related work**. Existing unpaired translation models, e.g., CycleGAN (Zhu et al., 2017a) or UNIT (Liu et al., 2017), typically have complex adversarial optimization objectives endowed with additional losses. These models require simultaneous optimization of several neural networks. Importantly, vanilla CycleGAN searches for a random translation map and is not capable of preserving certain attributes, e.g., the color, see (Lu et al., 2019, Figure 5b). To handle this issue, imposing extra losses is required (Benaim & Wolf, 2017; Kim et al., 2017), which further complicates the hyperparameter selection. In contrast, our approach has a straightforward objective (14); we use only 2 networks (potential $f$, map $T$), see Table 2 for the *comparison of hyperparameters*. While the majority of existing unpaired translation models are based on GANs, recent work (Su et al., 2023) proposes a diffusion model (DDIBs) and relates it to Schrödinger Bridge (Léonard, 2014), i.e., entropic OT.

## 5.3 ONE-TO-MANY TRANSLATION WITH OPTIMAL PLANS

We learn stochastic OT maps between various pairs of datasets for the $\gamma$-weak quadratic cost. The parameter $\gamma$ equals $\frac{2}{3}$ or $1$ in the experiments. We provide the results in Figures 1b and 6. In all the

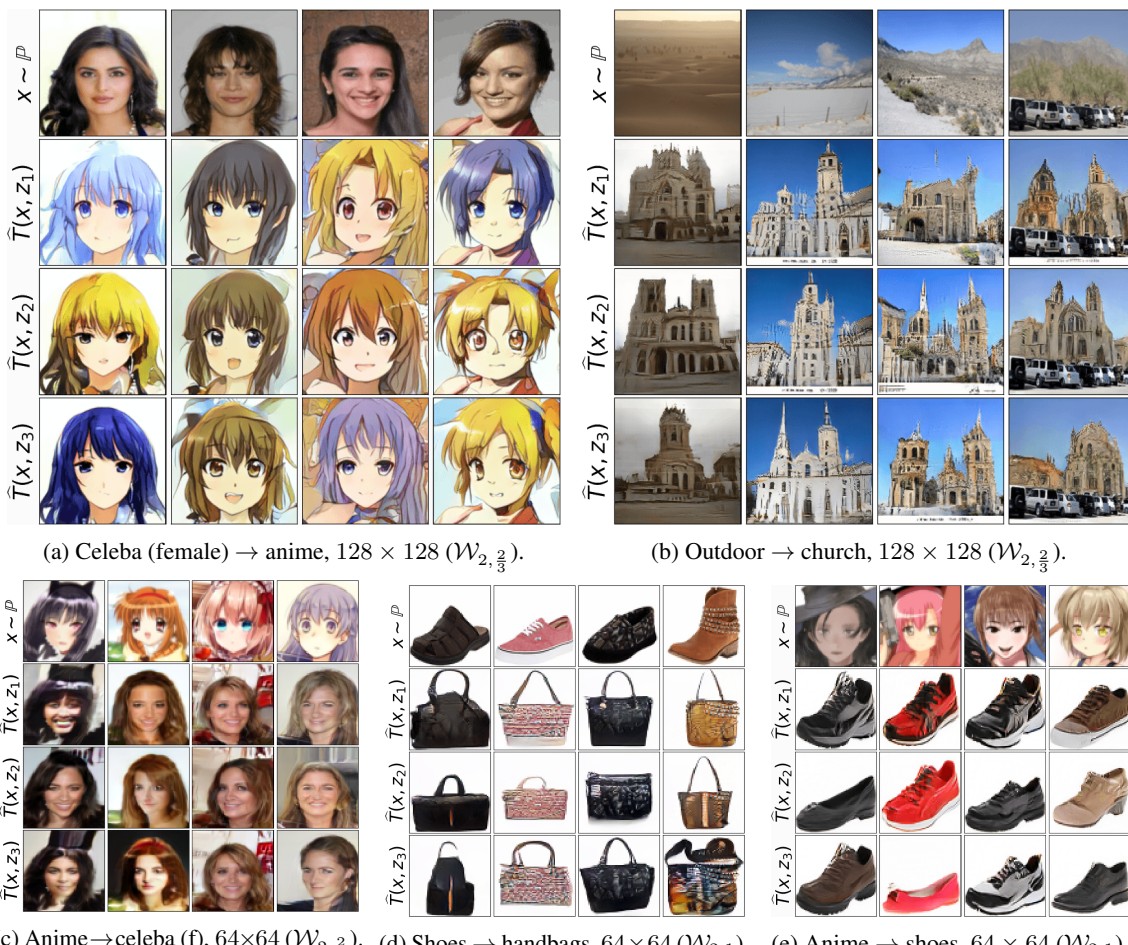

(a) Celeba (female) → anime, $128 \times 128$ ($\mathcal{W}_{2,\frac{2}{3}}$).

(b) Outdoor → church, $128 \times 128$ ($\mathcal{W}_{2,\frac{2}{3}}$).

(c) Anime→celeba (f), $64\times64$ ($\mathcal{W}_{2,\frac{2}{3}}$).  (d) Shoes → handbags, $64\times64$ ($\mathcal{W}_{2,1}$).   (e) Anime → shoes, $64 \times 64$ ($\mathcal{W}_{2,1}$).

Figure 6: Unpaired translation with stochastic OT maps ($\mathcal{W}_{2,\gamma}$).

cases, the random noise inputs $z \sim \mathbb{S}$ are not synchronized for different inputs $x$. The examples with the synchronized noise inputs $z$ are given in Appendix I. Extended results and examples of *interpolation* in the conditional latent space are given in Appendix H. The stochastic map $\widehat{T}(x, z)$ preserves the attributes of the input image and produces multiple outputs.

**Related work**. Transforming a one-to-one learning pipeline to one-to-many is nontrivial. Simply adding additional noise input leads to conditional collapse (Zhang, 2018). This is resolved by AugCycleGAN (Almahairi et al., 2018) and M-UNIT (Huang et al., 2018), but their optimization objectives are much more complicated then vanilla versions. Our method optimizes only 2 nets $f, T$ in straightforward objective (14). It offers a *single parameter* $\gamma$ to control the amount of variability in the learned maps. We refer to Table 2 for the comparison of *hyperparameters* of the methods.

## 6 DISCUSSION

**Potential impact.** Our method is a novel generic tool to align probability distributions with deterministic and stochastic transport maps. Beside unpaired translation, we expect our approach to be applied to other one-to-one and one-to-many unpaired learning tasks as well (image restoration, domain adaptation, etc.) and improve existing models in those fields. Compared to the popular models based on GANs (Goodfellow et al., 2014) or diffusion models (Ho et al., 2020), our method provides better interpretability of the learned map and allows to control the amount of diversity in generated samples (Appendix A). It should be taken into account that OT maps we learn might be suitable not for all unpaired tasks. We mark designing task-specific transport costs as a promising research direction.

**Limitations.** Our method searches for a solution $(f^*, T^*)$ of a saddle point problem (14) and extracts the stochastic OT map $T^*$ from it. We highlight after Lemma 4 and in §5.1 that not all $T^*$ are optimal stochastic OT maps. For strong costs, the issue leads to the conditional collapse. Studying saddle points of (14) and $\arg\inf$ sets (16) is an important challenge to address in the further research.

**Potential societal impact**. Our developed method is at the junction of optimal transport and generative learning. In practice, generative models and optimal transport are widely used in entertainment (image-manipulation applications like adding masks to images, hair coloring, etc.), design, computer graphics, rendering, etc. Our method is potentially applicable to many problems appearing in mentioned industries. While the mentioned applications allow making image processing methods publicly available, a potential negative is that they might transform some jobs in the graphics industry.

**Reproducibility.** We provide the source code for all experiments and release the checkpoints for all models of §5. The details are given in README.MD in the official repository.

ACKNOWLEDGEMENTS. The work was supported by the Analytical center under the RF Government (subsidy agreement 000000D730321P5Q0002, Grant No. 70-2021-00145 02.11.2021).

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

## A    VARIANCE-SIMILARITY TRADE-OFF

In this section, we study the effect of the parameter $\gamma$ on the structure of the learned stochastic map for the $\gamma$-weak quadratic cost. We consider *handbags $\rightarrow$ shoes* translation ($64 \times 64$) and test $\gamma \in \{0, \frac{1}{3}, \frac{2}{3}, 1\}$. The results are shown in Figure 7.

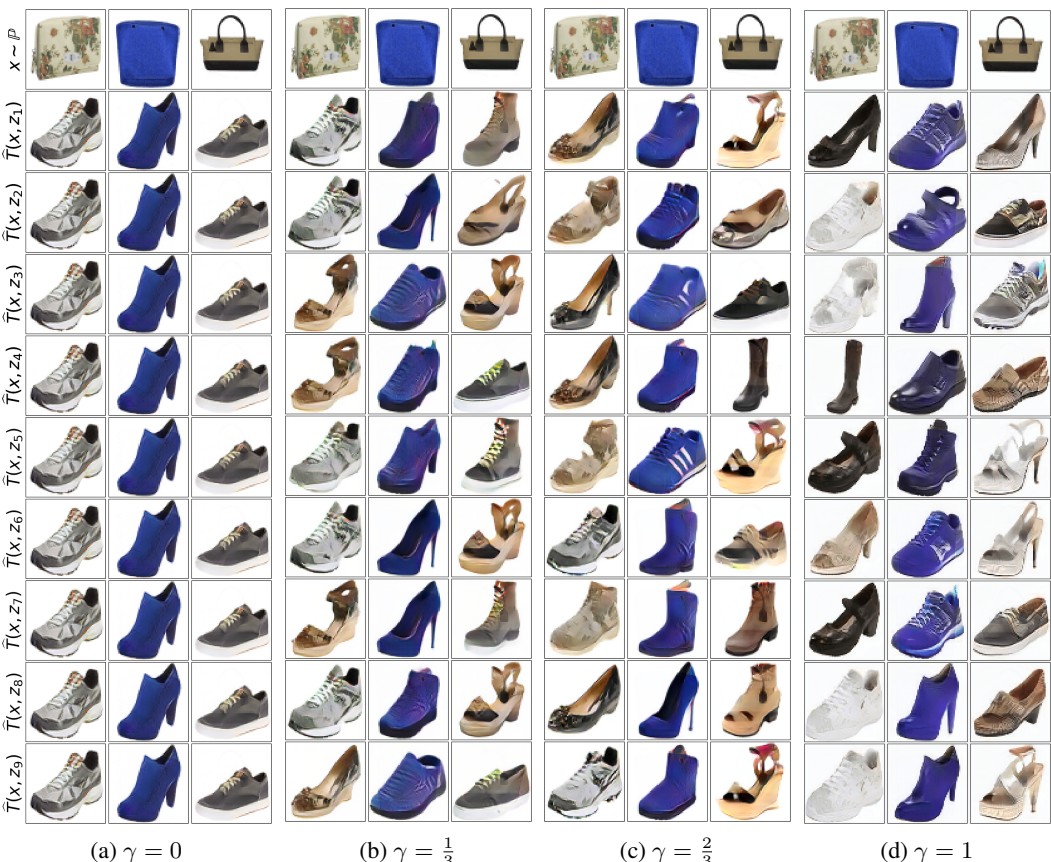

(a) $\gamma = 0$      (b) $\gamma = \frac{1}{3}$      (c) $\gamma = \frac{2}{3}$      (d) $\gamma = 1$

Figure 7: Stochastic *Handbags $\rightarrow$ shoes* translation with the $\gamma$-weak quadratic cost for various $\gamma$.

**Discussion.** For $\gamma = 0$ there is no variety in produced samples (Figure 7a), i.e., the conditional collapse happens. With the increase of $\gamma$ (Figures 7b, 7c), the variety of samples increases and the style of the input images is mostly preserved. For $\gamma = 1$ (Figure 7d), the variety of samples is very high but many of them do not preserve the style of the input image. The parameter $\gamma$ can be viewed as the **trade-off** parameter *balancing* the *variance* of samples and their **similarity** to the input.

## B    TOY 2D EXPERIMENTS

In this section, we test our Algorithm 1 on toy 2D distributions $\mathbb{P}, \mathbb{Q}$, i.e., $P = Q = 2$.

**Strong quadratic cost ($\gamma = 0$).** As we noted in §5.1 and Appendix A, for the strong quadratic cost, our method tends to learn deterministic maps $T(x, z) = T(x)$ which are independent of the noise input $z$. For deterministic maps $T(x)$, our method yields $\lfloor$MM:R$\rceil$ method which has been evaluated in the recent Wasserstein-2 benchmark by (Korotin et al., 2021b). The authors show that the method recovers OT maps well on synthetic high-dimensional pairs $\mathbb{P}, \mathbb{Q}$ with known ground truth OT maps. Thus, for brevity, we do not include toy experiments with our method for the strong quadratic cost.

**Weak quadratic cost ($\gamma > 0$).** To our knowledge, our method is the first to solve weak OT, i.e., there are no approaches to compare with. The analysis of computed transport plans for weak costs is challenging due to the lack of nontrivial pairs $\mathbb{P}, \mathbb{Q}$ with known ground truth OT plan $\pi^*$. The

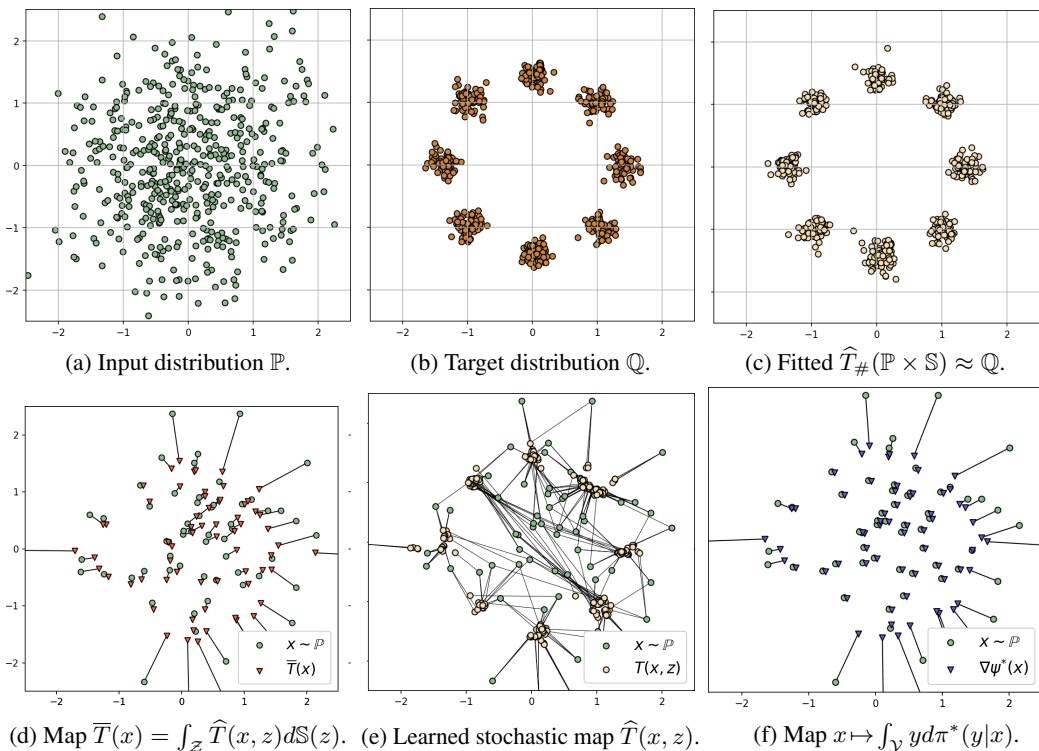

(a) Input distribution $\mathbb{P}$. (b) Target distribution $\mathbb{Q}$. (c) Fitted $\widehat{T}_\#(\mathbb{P} \times \mathbb{S}) \approx \mathbb{Q}$.

(d) Map $\overline{T}(x) = \int_{\mathcal{Z}} \widehat{T}(x, z) d\mathbb{S}(z)$. (e) Learned stochastic map $\widehat{T}(x, z)$. (f) Map $x \mapsto \int_{\mathcal{Y}} y \, d\pi^*(y|x)$.

Figure 8: *Gaussian → Mixture of 8 Gaussians*, learned stochastic map for the 1-weak quadratic cost.

situation is even worsened by the **nonuniqueness** of $\pi^*$. To cope with this issue, we consider the weak quadratic cost with $\gamma = 1$. For this cost, one may derive

$$C(x, \mu) = \int_{\mathcal{Y}} \frac{1}{2} \|x - y\|^2 d\mu(y) - \frac{1}{2}\text{Var}(\mu) = \frac{1}{2}\|x - \int_{\mathcal{Y}} y \, d\mu(y)\|^2. \tag{21}$$

For cost (21) and a pair $\mathbb{P}, \mathbb{Q}$, (Gozlan & Juillet, 2020, Theorem 1.2) states that there exists a $\mathbb{P}$-unique (up to a constant) convex $\psi : \mathbb{R}^P \to \mathbb{R}$ such that every OT plan $\pi^*$ satisfies $\nabla\psi(x) = \int_{\mathcal{Y}} y \, d\pi^*(y|x)$. Besides, $\nabla\psi : \mathbb{R}^P \to \mathbb{R}^P$ is 1-Lipschitz. Let $\widehat{T}(x, z)$ be the stochastic map recovered by our Algorithm 1, and let $\widehat{\pi}$ be the corresponding plan. Let

$$\overline{T}(x) \stackrel{def}{=} \int_{\mathcal{Y}} y \, d\widehat{\pi}(y|x) = \int_{\mathcal{Z}} \widehat{T}(x, z) d\mathbb{S}(z). \tag{22}$$

Due to the above mentioned characterization of OT plans, $\overline{T}(x)$ should look like a gradient $\nabla\psi(x)$ of some convex function $\psi(x)$ and should nearly be a contraction. Since here we work in the 2D space, we are able to get sufficiently many samples from $\mathbb{P}$ and $\mathbb{Q}$ and obtain a fine approximation of an OT plan $\pi^*$ and $\nabla\psi$ by a discrete weak OT solver. We may sample random batches from $X \sim \mathbb{P}$ and $Y \sim \mathbb{Q}$ of size $2^{10}$ and use `ot.weak` from POT library[4] to get some optimal $\pi^*$ and $\nabla\psi = \int_{\mathcal{Y}} y \, d\pi^*(y|x)$. We are going to compare our recovered average map $\overline{T}$ with $\nabla\psi$.

**Datasets.** We test 2 pairs $\mathbb{P}, \mathbb{Q}$: *Gaussian → Mixture of 8 Gaussians*; *Gaussian → Swiss roll*.

**Neural Networks.** We use multi-layer perceptrons as $f_\omega, T_\theta$ with 3 hidden layers of 100 neurons and ReLU nonlinearity. The input of the stochastic map $T_\theta(x, z)$ is $2 + 2 = 4$ dimensional. The two first dimensions represent the input $x \in \mathbb{R}^2$ while the other dimensions represent the noise $z \sim \mathbb{S}$. We employ a Gaussian noise with $\sigma = 0.1$

**Discussion.** We provide qualitative results in Figures 8 and 9. In both cases, the pushforward distribution $\widehat{T}_\#(\mathbb{P} \times \mathbb{S})$ matches the desired target distribution $\mathbb{Q}$ (Figures 8c and 9c). Figures 8e

---

[4]https://pythonot.github.io/

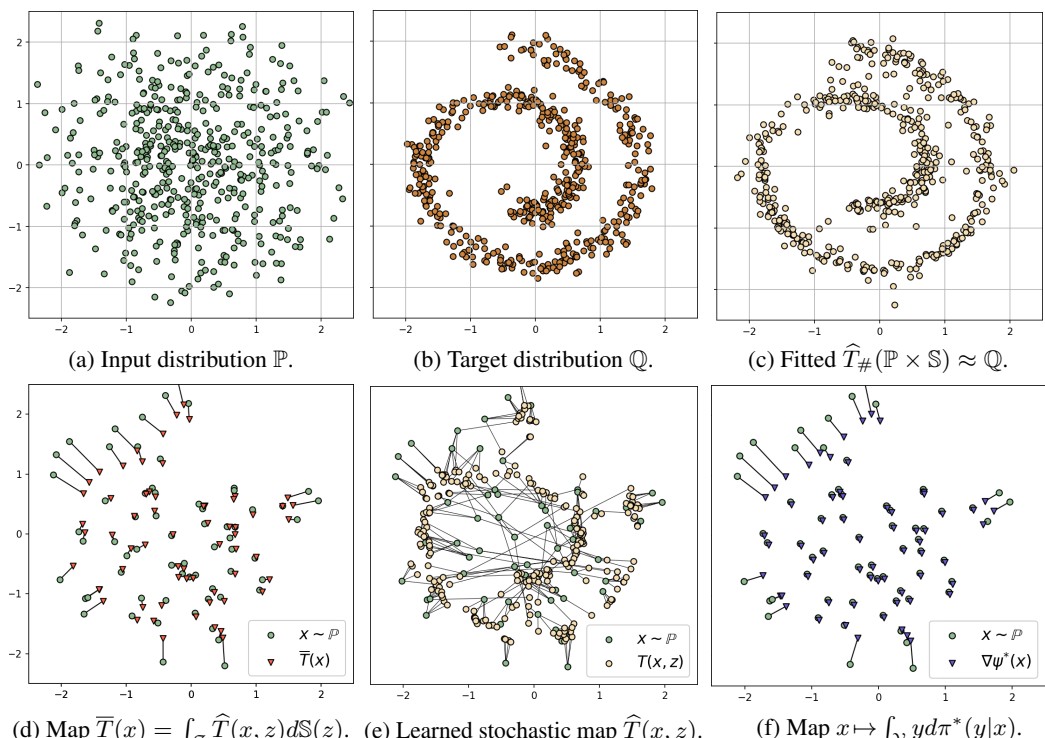

(a) Input distribution $\mathbb{P}$.     (b) Target distribution $\mathbb{Q}$.     (c) Fitted $\widehat{T}_{\#}(\mathbb{P} \times \mathbb{S}) \approx \mathbb{Q}$.

(d) Map $\overline{T}(x) = \int_{\mathcal{Z}} \widehat{T}(x, z) d\mathbb{S}(z)$.    (e) Learned stochastic map $\widehat{T}(x, z)$.    (f) Map $x \mapsto \int_{\mathcal{Y}} y d\pi^*(y|x)$.

Figure 9: *Gaussian → Swiss Roll*, learned stochastic OT map for the 1-weak quadratic cost.

and 9e show how the mass of points $x \sim \mathbb{P}$ is split by the stochastic map. The average maps $\overline{T}(x)$ (Figures 8d, 9d) indeed nearly match the ground truth $\nabla\psi$ (Figures 8f, 9f) obtained by POT. To quantify them, we compute $\mathcal{L}^2$-UVP $(\overline{T}) = 100\% \cdot \|\overline{T} - \nabla\psi\|_{\mathbb{P}}^2 / \text{Var}(\nabla\psi_{\#}\mathbb{P})$ metric (Korotin et al., 2021a, §5.1). Here we obtain small values $< 1\%$ and $\approx 3\%$ for the Swiss Roll and 8 Gaussians examples which further indicates the similarity of the learned $\overline{T}$ and the ground truth $\nabla\psi(x)$.

Note that $\overline{T}$ indeed roughly equals a gradient of a convex function. The gradients of convex functions are cycle monotone (Rockafellar, 1966). Cycle monotonicity yields that for $x_1 \neq x_2$ the segments $[x_1, \nabla\psi(x_1)]$ and $[x_2, \nabla\psi(x_2)]$ do not intersect in the inner points (Villani, 2008, §8).[5] Visually, we see that in Figures 8d and 9d the segments $[x, \overline{T}(x)]$ do not intersect for different $x$, which is good.

## C   Toy 1D Experiments

In this section, we additionally test our Algorithm 1 on toy 1D distributions $\mathbb{P}, \mathbb{Q}$, i.e., $P = Q = 1$. In this case, transport plans are 2D distributions and can be conveniently visualized.

We experiment with the 1-weak quadratic cost (21). Following the discussion in the previous section, we recall that an OT plan $\pi^*$ may be not unique. However, all OT plans satisfy $\nabla\psi(x) = \int_{\mathcal{Y}} y \, d\pi^*(y|x)$ for some 1-smooth convex function $\psi : \mathbb{R} \to \mathbb{R}$. This simply means that $x \mapsto \nabla\psi(x) = \int_{\mathcal{Y}} y \, d\pi^*(y|x)$ is a monotone increasing 1-Lipschitz function $\nabla\psi : \mathbb{R} \to \mathbb{R}$. Below we check that this necessary condition holds for $\overline{T}$ (22), where $\widehat{T}$ is our learned stochastic map.

**Datasets.** We test 2 pairs $\mathbb{P}, \mathbb{Q}$: *Gaussian → Mix of 2 Gaussians*; *Gaussian → Mix of 3 Gaussians*.

**Neural Networks.** We use the same networks as in Appendix B. This time, the input of the stochastic map $T_\theta(x, z)$ is $1 + 1 = 2$ dimensional, the input to $f_\omega - 1$-dimensional.

**Discussion.** We provide qualitative results in Figures 10 and 11. For each case, we plot the results of 3 random restarts of our method ($\hat{\pi}$ denotes our learned OT plan). Similarly to Appendix B, we plot

---

[5]For the sake of clarity, we slightly reformulated the property of the cycle monotone maps (Villani, 2008).

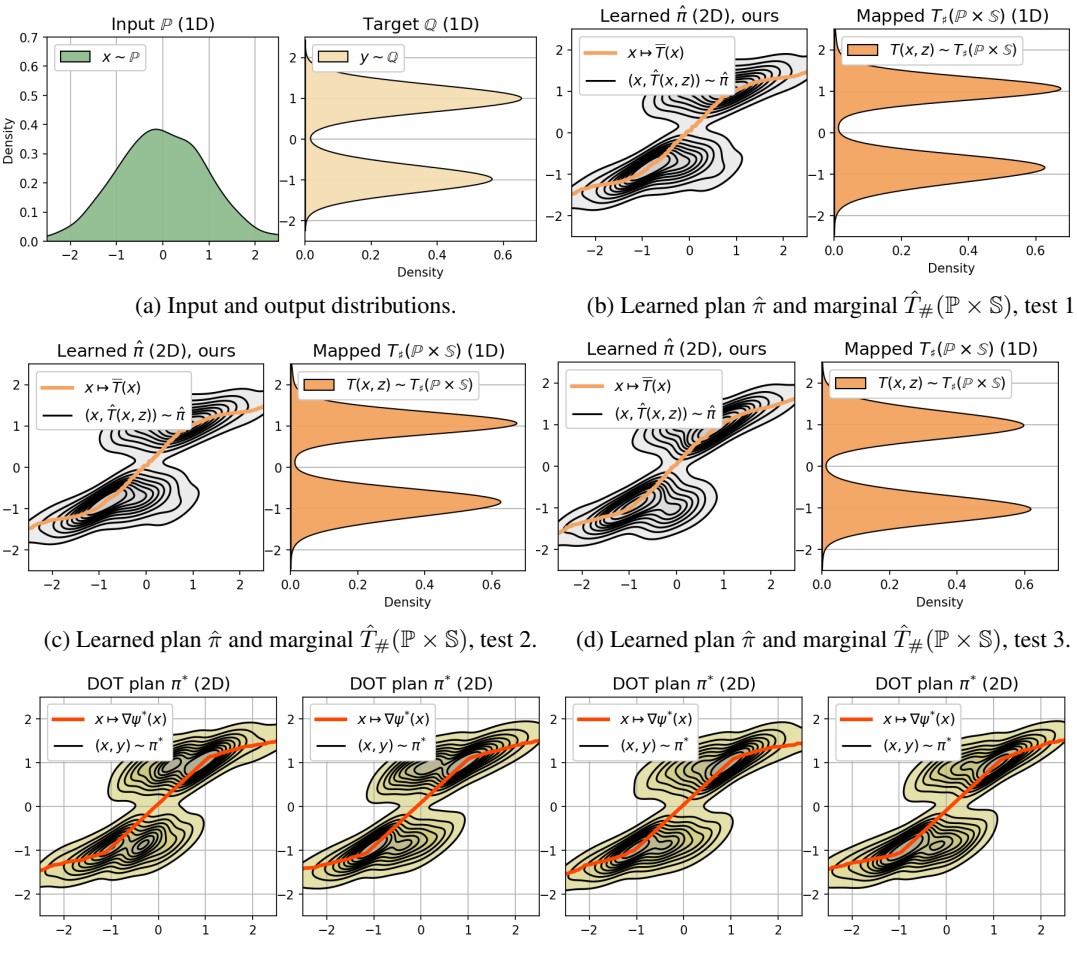

(a) Input and output distributions.

(b) Learned plan $\hat{\pi}$ and marginal $\hat{T}_{\#}(\mathbb{P} \times \mathbb{S})$, test 1.

(c) Learned plan $\hat{\pi}$ and marginal $\hat{T}_{\#}(\mathbb{P} \times \mathbb{S})$, test 2.

(d) Learned plan $\hat{\pi}$ and marginal $\hat{T}_{\#}(\mathbb{P} \times \mathbb{S})$, test 3.

(e) Various optimal plans $\pi^*$ learned by discrete OT (considered here as the ground truth).

Figure 10: Stochastic plans between toy 1D distributions (Figure 10a) learned by NOT (Figures 10b, 10c, 10d) and discrete OT (Figure 10e) with the 1-weak quadratic cost. The figures with the 2D transport plans also demonstrate the average map $x \mapsto \int_{\mathcal{Y}} y d\hat{\pi}(y|x)$ (conditional expectation).

the results obtained by a discrete weak OT solver (`ot.weak` from POT library). Namely, in Figures 10e, 11e we show its results obtained for 4 restarts with differing seeds. Note that the average maps $\overline{T}$ computed by our algorithm in both cases nearly match those computed by the discrete weak OT. This indicates that the transport cost of our computed plan $\hat{\pi}$ is since

$$[\text{Cost of } \hat{\pi}] = \int_{\mathcal{X}} \frac{1}{2} \|x - \underbrace{\overline{T}(x)}_{\approx \psi(x)}\|^2 d\mathbb{P}(x) \approx \int_{\mathcal{X}} \frac{1}{2} \|x - \nabla\psi(x)\|^2 d\mathbb{P}(x) = \text{Cost}(\mathbb{P}, \mathbb{Q}),$$

i.e., it nearly equals the optimal cost. Here we use $\overline{T}(x) \approx \nabla\psi(x)$ observed from the experiments. To conclude, wee see that the recovered plans are close to the DOT considered as the ground truth.

## D  COMPARISON WITH PRINCIPAL UNPAIRED TRANSLATION METHODS

We compare our Algorithm 1 with popular models for unpaired translation. We consider *handbags* → *shoes* ($64 \times 64$), *celeba male* → *female* ($64 \times 64$), *outdoor* → *church* ($128 \times 128$) translation. For quantitative comparison, we compute Frechet Inception Distance[6] (Heusel et al., 2017, FID) of the

---

[6]`github.com/mseitzer/pytorch-fid`

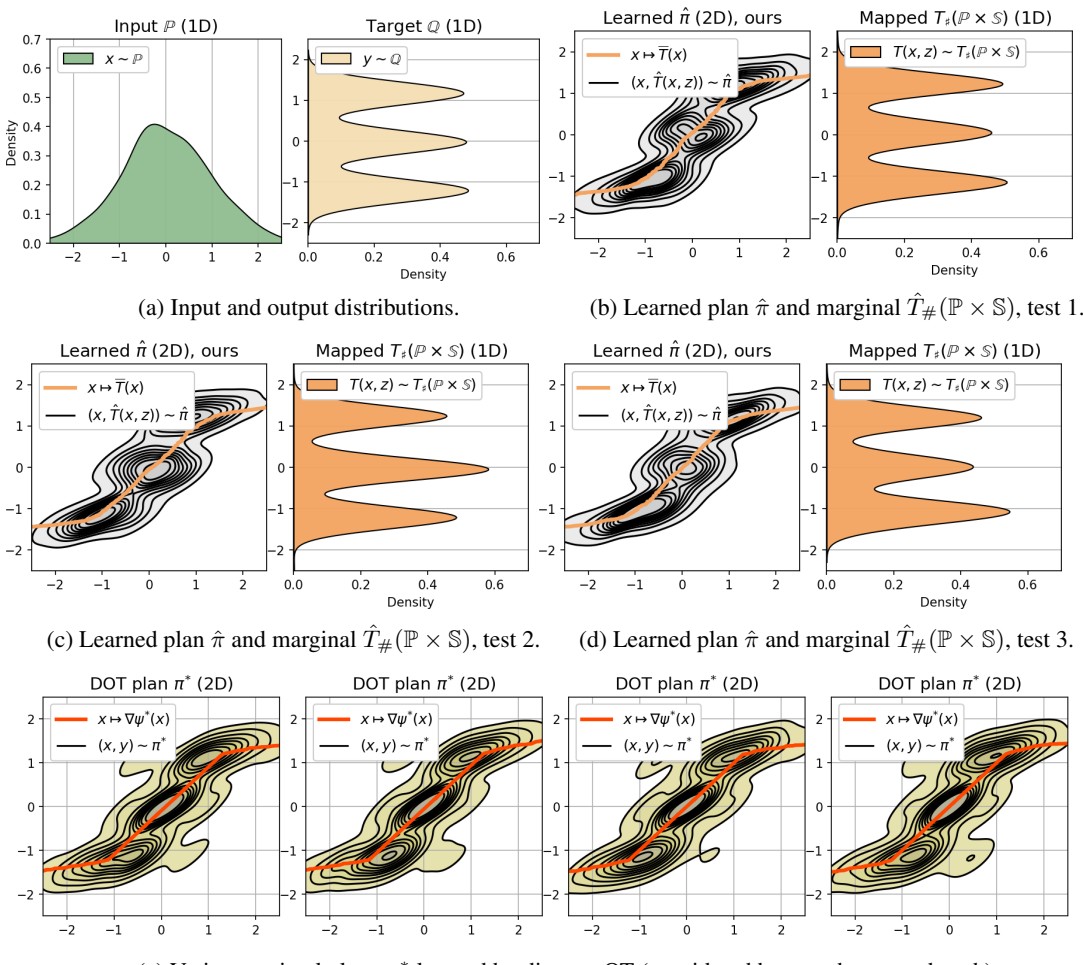

(a) Input and output distributions.

(b) Learned plan $\hat{\pi}$ and marginal $\hat{T}_{\#}(\mathbb{P} \times \mathbb{S})$, test 1.

(c) Learned plan $\hat{\pi}$ and marginal $\hat{T}_{\#}(\mathbb{P} \times \mathbb{S})$, test 2.

(d) Learned plan $\hat{\pi}$ and marginal $\hat{T}_{\#}(\mathbb{P} \times \mathbb{S})$, test 3.

(e) Various optimal plans $\pi^*$ learned by discrete OT (considered here as the ground truth).

Figure 11: Stochastic plans between toy 1D distributions (Figure 11a) learned by NOT (Figures 11b, 11c, 11d) and discrete OT (Figure 11e) with the 1-weak quadratic cost. The figures with the 2D transport plans also demonstrate the average map $x \mapsto \int_{\mathcal{Y}} y d\hat{\pi}(y|x)$ (conditional expectation).

mapped **test** handbags subset w.r.t. the **test** shoes subset. The scores of our method and alternatives are given in Table 1. The translated images are shown in Figures 12, 13, 14.

**Methods.** We compare our method with one-to-one CycleGAN [7](Zhu et al., 2017a), DiscoGAN[8] (Kim et al., 2017) and with one-to-many AugCycleGAN[9] (Almahairi et al., 2018) and MUNIT[10] (Huang et al., 2018). We use the official or community implementations with the hyperparameters from the respective papers. We choose the above-mentioned methods for comparison because they are *principal* methods for one-to-one and one-to-many translation. Recent methods (GMM-UNIT (Liu et al., 2020), COCO-FUNIT (Saito et al., 2020), StarGAN (Choi et al., 2020)) are based on them and focus on specific details/setups such as style/content separation, few-shot learning, disentanglement, multi-domain transfer, which are *out of scope* of our paper.

**Discussion.** Existing one-to-one methods visually preserve the style during translation comparably to our method. Alternative one-to-many methods do not preserve the style at all. When the input

---

[7]github.com/eriklindernoren/PyTorch-GAN/tree/master/implementations/cyclegan

[8]github.com/eriklindernoren/PyTorch-GAN/tree/master/implementations/discogan

[9]github.com/aalmah/augmented_cyclegan

[10]github.com/NVlabs/MUNIT

and output domains are similar (*handbags→shoes*, *celeba male → female*), the FID scores of all the models are comparable. However, most models are outperformed by NOT when the domains are distant (*outdoor → church*), see Figure 14 and the last row in Table 1. For completeness, in Table 2 we compare the number of hyperparameters of the translation methods in view. Note that in contrast to the other methods, we optimize only 2 neural networks – transport map and potential.

| Type | One-to-one | | | One-to-many | | |
|---|---|---|---|---|---|---|
| **Method** | Disco GAN | Cycle GAN | NOT (ours) | AugCycle GAN | MUNIT | NOT (ours) |
| Handbags → shoes (64× 64) | 22.42 | 16.00 | **13.77** | 18.84 ± 0.11 | 15.76 ± 0.11 | **13.44** ± 0.12 |
| Celeba male → female (64× 64) | 35.64 | 17.74 | **13.23** | 12.94 ±0.08 | 17.07 ±0.11 | **11.96** ±0.07 |
| Outdoor → church (128× 128) | 75.36 | 46.39 | **25.5** | 51.42 ±0.12 | 31.42 ±0.16 | **25.97** ±0.14 |

Table 1: Test FID↓ of the considered unpaired translation methods.

| Type | One-to-one | | | One-to-many | | |
|---|---|---|---|---|---|---|
| **Method** | Disco GAN | Cycle GAN | NOT (ours) | AugCycle GAN | MUNIT | NOT (ours) |
| Hyperparameters of optimization objectives | None | Weights of cycle and identity losses $\lambda_{cyc}, \lambda_{id}$ | None | Weights of cycle losses $\gamma_1, \gamma_2$ | Weights of reconstruction losses $\lambda_x, \lambda_c, \lambda_s$ | Diversity control parameter $\gamma$ |
| Total number of hyperparameters | **0** | 2 | **0** | 2 | 3 | **1** |
| Networks | 2 generators, 2×29.2M 2 discriminators 2×0.7M | 2 generators 2×11.4M 2 discriminators 2×2.8M | 1 transport 9.7M, 1 potential 22.9M [32.4M*] | 2 generators 2×1.1M, 2 discriminators 2×2.8M, 2 encoders 2×1.4M | 2 generators 2×15.0M, 2 discriminators 2×8.3M | 1 transport map 9.7M, 1 potential 22.9M [32.4M*] |
| Total number of networks and parameters | 4 networks 59.8M | 4 networks 28.2M | **2 networks** 32.6M [42.1M*] | 6 networks 7.0M | 4 networks 46.6M | **2 networks** 32.6M [42.1M*] |

Table 2: Comparison of the number of hyperparameters of the optimization objectives, the number of networks and their parameters for the considered unpaired translation methods for 64×64 images.

---

* For 128 × 128 images.

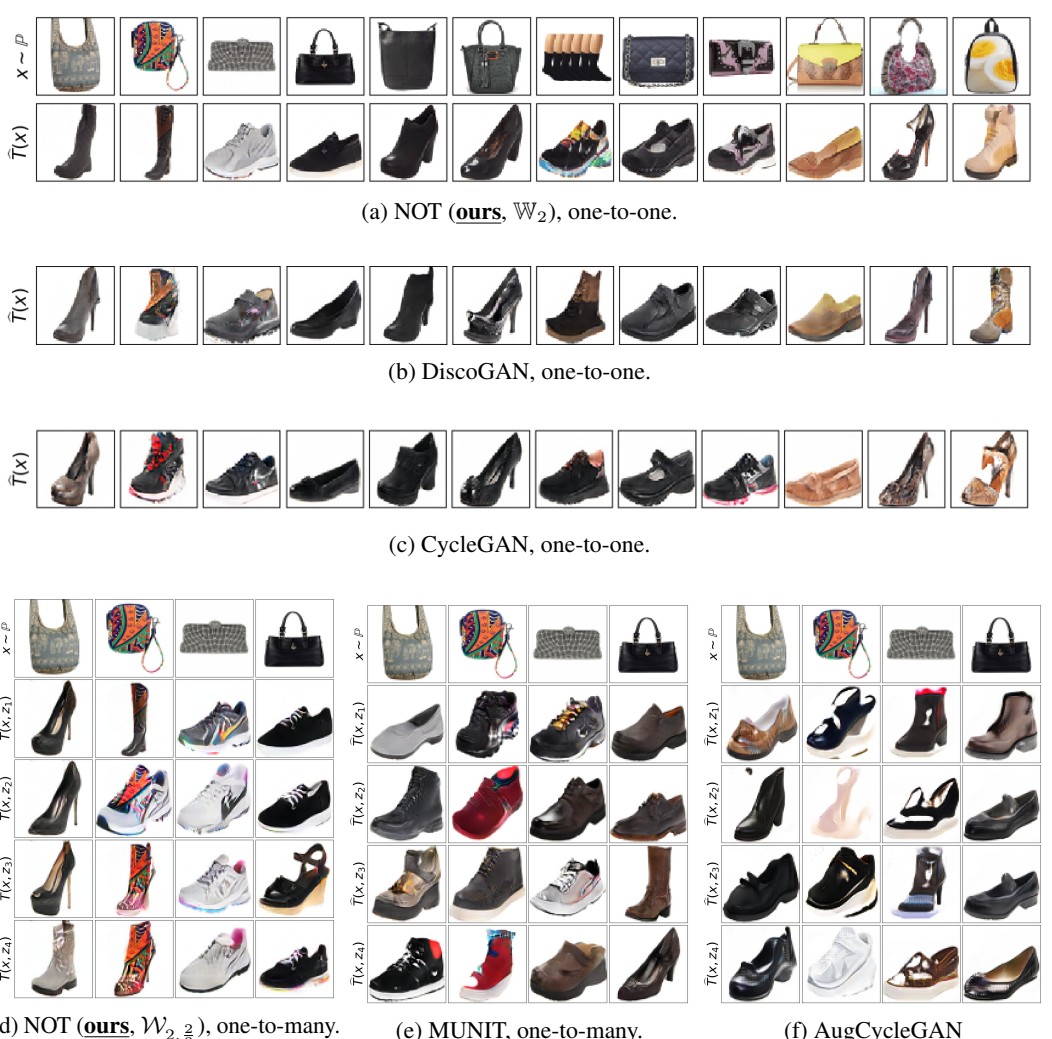

Figure 12: *Handbags → shoes* translation ($64 \times 64$) by the methods in view.

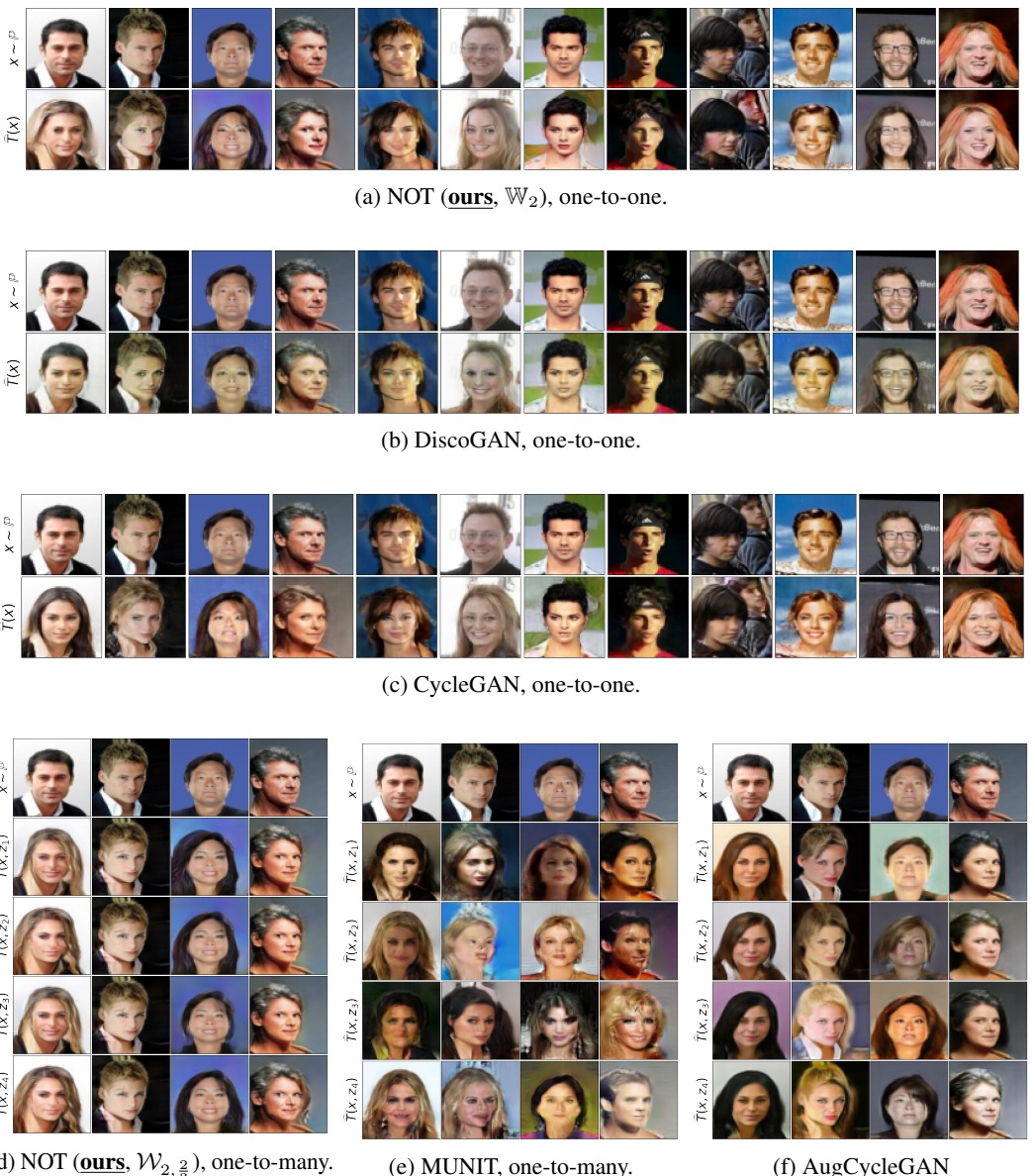

Figure 13: *Celeba (male)* → *Celeba (female)* translation ($64 \times 64$) by the methods in view.

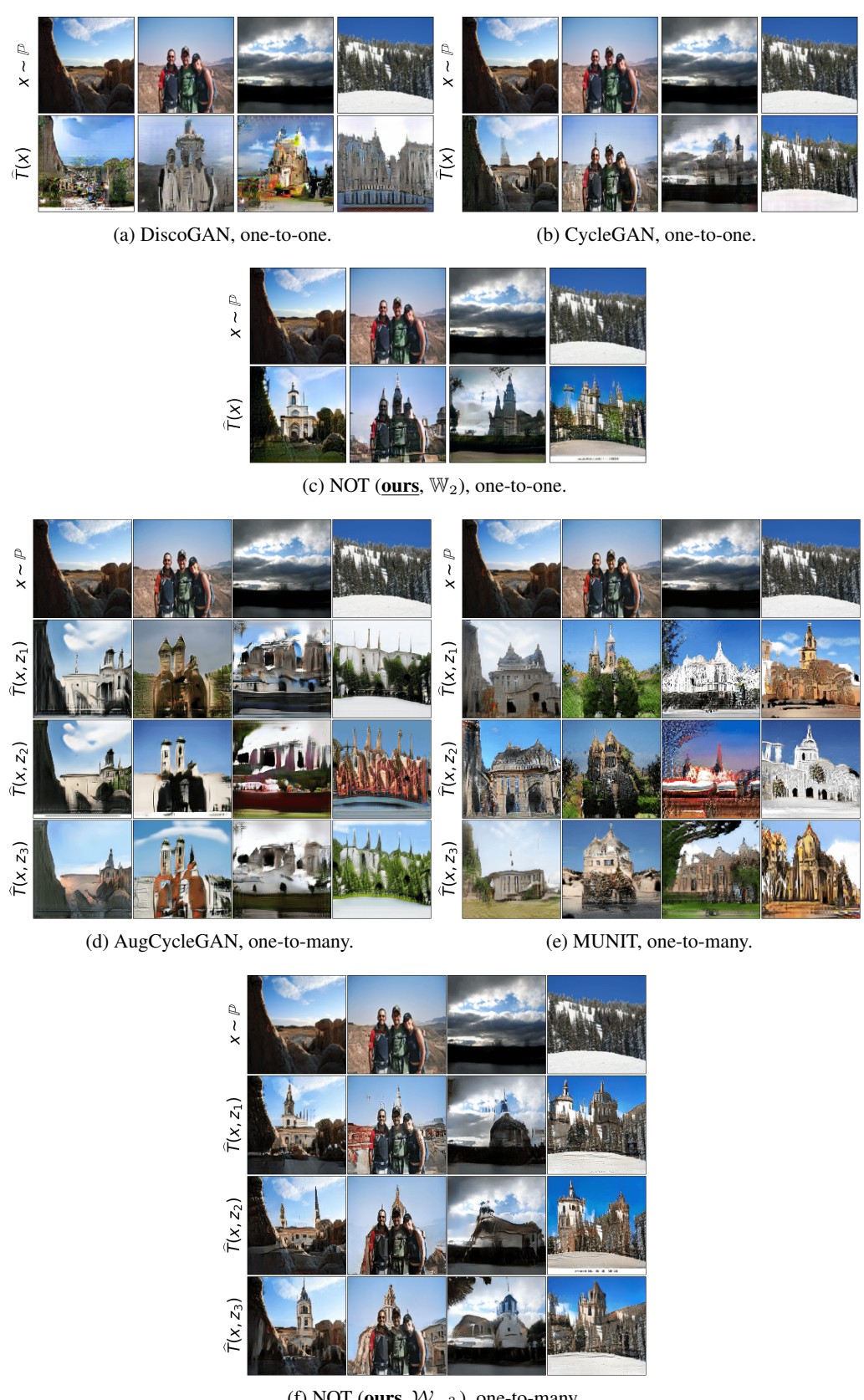

Figure 14: *Outdoor → church* ($128 \times 128$) translation with various methods.

# E EXPERIMENTAL DETAILS

**Pre-processing.** We beforehand rescale anime face images to $512 \times 512$, and do $256 \times 256$ crop with the center located 14 pixels above the image center to get the face. Next, for all these datasets, we rescale RGB channels to $[-1, 1]$ and resize images to the required size ($64 \times 64$ or $128 \times 128$). We do not apply any augmentations to data.

**Neural networks.** We use WGAN-QC discriminator's ResNet architecture (Liu et al., 2019) for potential $f$. We use UNet (Ronneberger et al., 2015) as the stochastic transport map $T(x, z)$. The noise $z$ is simply an additional 4th input channel (RGBZ), i.e., the dimension of the noise equals the image size ($64 \times 64$ or $128 \times 128$). We use high-dimensional Gaussian noise with axis-wise $\sigma = 0.1$.

**Optimization.** We use the Adam optimizer (Kingma & Ba, 2014) with the default betas for both $T_\theta$ and $f_\omega$. The learning rate is $lr = 1 \cdot 10^{-4}$. The batch size is $|X| = 64$. The number of inner iterations is $k_T = 10$. When training with the weak cost (4), we sample $|Z_x| = 4$ noise samples per each image $x$ in batch. In toy experiments, we do 10K total iterations of $f_\omega$ update. In the experiments with unpaired translation, our Algorithm 1 converges in $\approx 40$K iterations for most datasets.

**Dynamic weak cost.** In §5.3, we train the algorithm with the gradually changing $\gamma$. Starting from $\gamma = 0$, we linearly increase it to the desired value ($\frac{2}{3}$ or 1) during 25K first iterations of $f_\omega$.

**Stability of training.** In several cases, we noted that the optimization fluctuates around the saddle points or diverges. An analogous behavior of saddle point methods for OT has been observed in (Korotin et al., 2021b). For the $\gamma$-weak quadratic cost ($\gamma > 0$), we sometimes experienced instabilities when the input $\mathbb{P}$ is notably less disperse than $\mathbb{Q}$ or when the parameter $\gamma$ is high. Studying this behaviour and improving stability/convergence of the optimization is a promising research direction.

**Computational complexity.** The time and memory complexity of training deterministic OT maps $T(x)$ is comparable to that of training usual generative models for unpaired translation. Our networks converge in 1-3 days on a Tesla V100 GPU (16 GB); wall-clock times depend on the datasets and the image sizes. Training stochastic $T(x, z)$ is harder since we sample multiple random $z$ per $x$ (we use $|Z| = 4$). Thus, we learn stochastic maps on $4 \times$ Tesla V100 GPUs.

# F OPTIMALITY OF SOLUTIONS FOR STRICTLY CONVEX COSTS

Our Lemma 4 proves that optimal maps $T^*$ are contained in the $\arg\inf_T$ sets of optimal potentials $f^*$ but leaves the question what else may be contained in these $\arg\inf_T$ sets open. Our following result shows that for *strictly* convex costs, nothing else beside OT maps is contained there.

**Lemma 5** (Solutions of the maximin problem are OT maps). *Let $C(x, \mu)$ be a weak cost which is strictly convex in $\mu$. Assume that there exists at least one potential $f^*$ which maximizes dual form* (5). *Consider any such optimal potential $f^* \in \arg\sup_f \inf_T \mathcal{L}(f, T)$. It holds that*

$$\hat{T} \in \arg\inf_T \mathcal{L}(f^*, T) \Rightarrow \hat{T} \text{ is a stochastic OT map.}$$

*Proof of Lemma* (5). By the definition of $f^*$, we have

$$\mathcal{L}(f^*, \hat{T}) = \sup_f \inf_T \mathcal{L}(f, T) = \text{Cost}(\mathbb{P}, \mathbb{Q}),$$

i.e., $\hat{T}$ attains the optimal cost. It remains to check that it satisfies $\hat{T} \sharp (\mathbb{P} \times \mathbb{S}) = \mathbb{Q}$, i.e., $\hat{T}$ generates $\mathbb{Q}$ from $\mathbb{P}$. Let $T^*$ be any true stochastic OT map. We denote $\mu_x^* = T^*(x, \cdot) \sharp \mathbb{S}$ and $\hat{\mu}_x = \hat{T}(x, \cdot) \sharp \mathbb{S}$ for all $x \in \mathcal{X}$ and define $\mu_x^1 = \frac{1}{2}(\mu_x^* + \hat{\mu}_x)$. Let $T^1 : \mathcal{X} \times \mathcal{Z} \to \mathcal{Y}$ be any stochastic map which satisfies $T^1(x, \cdot) \sharp \mathbb{S} = \mu_x^1$ for all $x \in \mathcal{X}$ (§4.1). By using the change of variables, we derive

$$\text{Cost}(\mathbb{P}, \mathbb{Q}) \geq \mathcal{L}(f^*, T^1) = \int_{\mathcal{X}} C(x, \mu_x^1) d\mathbb{P}(x) - \int_{\mathcal{X}} \Big[ \int_{\mathcal{Y}} f^*(y) d\mu_x^1(y) \Big] d\mathbb{P}(x) + \int_{\mathcal{Y}} f^*(y) \mathbb{Q}(y). \tag{23}$$

---

github.com/milesial/Pytorch-UNet

Since $C$ is convex in the second argument, we have

$$C(x, \mu_1^x) = C\big(x, \frac{1}{2}(\mu_x^* + \hat{\mu}_x)\big) \geq \frac{1}{2}C(x, \mu_x^*) + \frac{1}{2}C(x, \hat{\mu}_x). \qquad (24)$$

Since $C$ is strictly convex, the equality in (24) is possible only when $\mu_x^* = \hat{\mu}_x$. We also note that

$$\int_{\mathcal{Y}} f^*(y)d\mu_x^1(y) = \int_{\mathcal{Y}} f^*(y)d\frac{(\mu_x^* + \hat{\mu}_x)}{2}(y) = \frac{1}{2}\int_{\mathcal{Y}} f^*(y)d\mu_x^*(y) + \frac{1}{2}\int_{\mathcal{Y}} f^*(y)d\hat{\mu}_x(y).$$

We substitute these findings to $\mathcal{L}(f^*, T^1)$ and get

$$\text{Cost}(\mathbb{P}, \mathbb{Q}) \geq \mathcal{L}(f^*, T^1) \geq \frac{1}{2}\mathcal{L}(f^*, T^*) + \frac{1}{2}\mathcal{L}(f^*, \hat{T}) = \frac{1}{2}\text{Cost}(\mathbb{P}, \mathbb{Q}) + \frac{1}{2}\text{Cost}(\mathbb{P}, \mathbb{Q}) = \text{Cost}(\mathbb{P}, \mathbb{Q}).$$

Thus, (23) is an equality $\mathbb{P}$-almost surely for all $x \in \mathcal{X}$ and $\mu_x^* = \hat{\mu}_x$ holds $\mathbb{P}$-almost surely. This means that $T^*$ and $\hat{T}$ generate the same distribution from $\mathbb{P} \times \mathbb{S}$, i.e., $\hat{T}$ is a stochastic OT map. $\quad\square$

Our generic framework allows learning stochastic transport maps (Lemma 4). For strictly convex costs, all the solutions of our objective (14) are guaranteed to be stochastic OT maps (Lemma 5). In the experiments, we focus on strong and weak *quadratic* costs, which are not strictly convex but still provide promising performance in the downstream task of unpaired image-to-image translation (§5). Developing strictly convex costs is a promising research avenue for the future work.

## G   RELATION TO PRIOR WORKS IN UNBALANCED OPTIMAL TRANSPORT

In the context of OT, (Yang & Uhler, 2019) employ a stochastic generator to learn a transport plan $\pi$ in the unbalanced OT problem (Chizat, 2017). Due to this, their optimization objective slightly resembles our objective (15). However, this similarity is deceptive. Unlike strong (2) or weak (3) OT, the unbalanced OT is an **unconstrained** problem, i.e., there is no need to satisfy $\pi \in \Pi(\mathbb{P}, \mathbb{Q})$. This makes unbalanced OT easier to handle: to optimize it one just has to parametrize the plan $\pi$ and backprop through the loss. The challenging part with which the authors deal is the estimation of the $\phi$-divergence terms in the unbalanced OT objective. These terms can be interpreted as a *soft relaxation* of the constraints $\pi \in \Pi(\mathbb{P}, \mathbb{Q})$, i.e., penalization for disobeying the constraints. The authors compute these terms by employing the variational (dual) formula from $f$-GAN (Nowozin et al., 2016). This yields a GAN-style optimization problem $\min_T \max_f$ which is **similar** to other problems in the generative adversarial framework. The problem we tackle is strong (2) and weak (3) OT which requires enforcing of the constraint $\pi \in \Pi(\mathbb{P}, \mathbb{Q})$. We reformulate the dual (weak) OT problem (5) into maximin problem (15) which can be used to recover the OT plan (via the stochastic map $T$). Our approach can be viewed as a *hard enforcement* of the constraints. Our $\max_f \min_T$ saddle point problem (15) is **atypical** for the traditional generative adversarial framework.

# H ADDITIONAL EXPERIMENTAL RESULTS

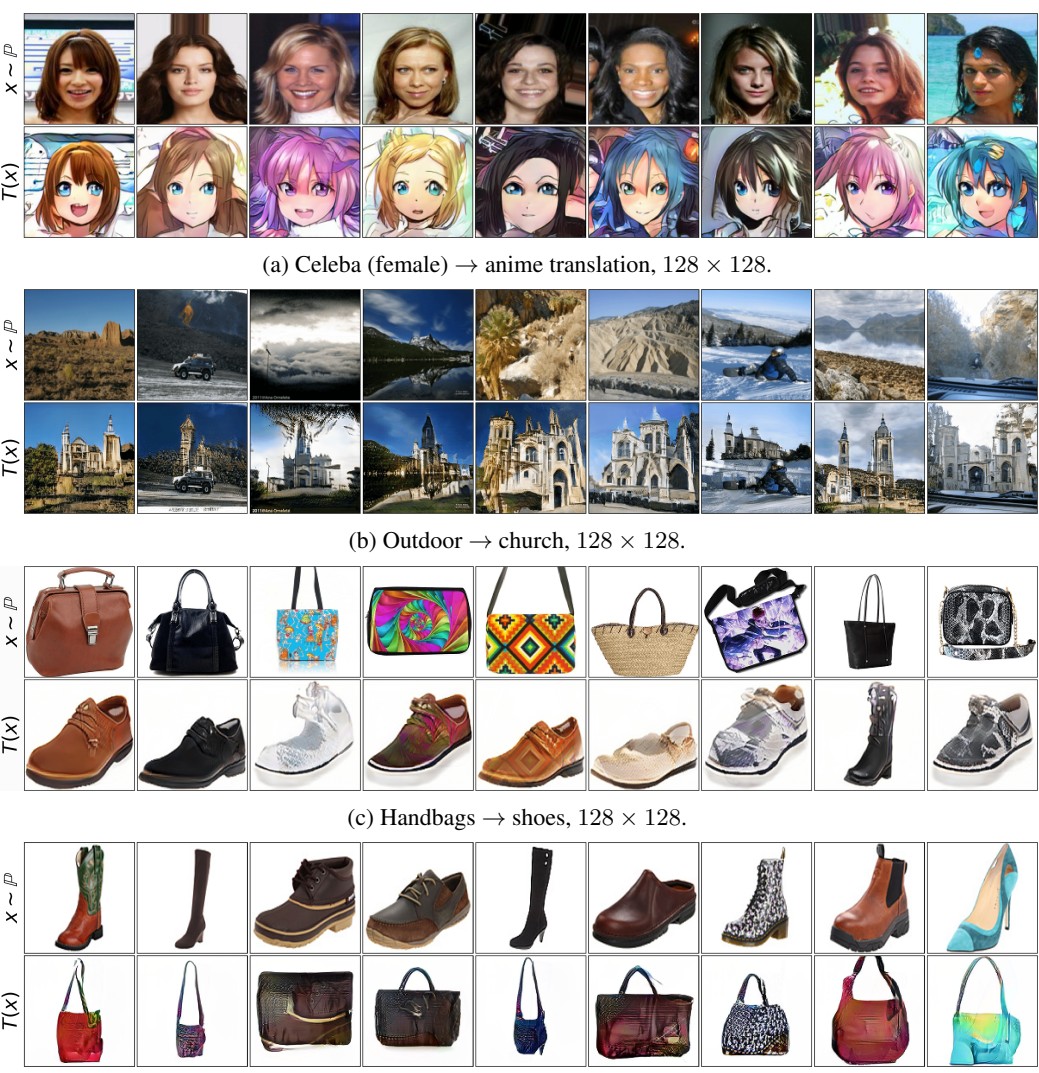

(a) Celeba (female) → anime translation, 128 × 128.

(b) Outdoor → church, 128 × 128.

(c) Handbags → shoes, 128 × 128.

(d) Shoes → handbags, 128 × 128.

Figure 15: Unpaired translation with OT maps ($\mathbb{W}_2$). Additional examples (part 1).

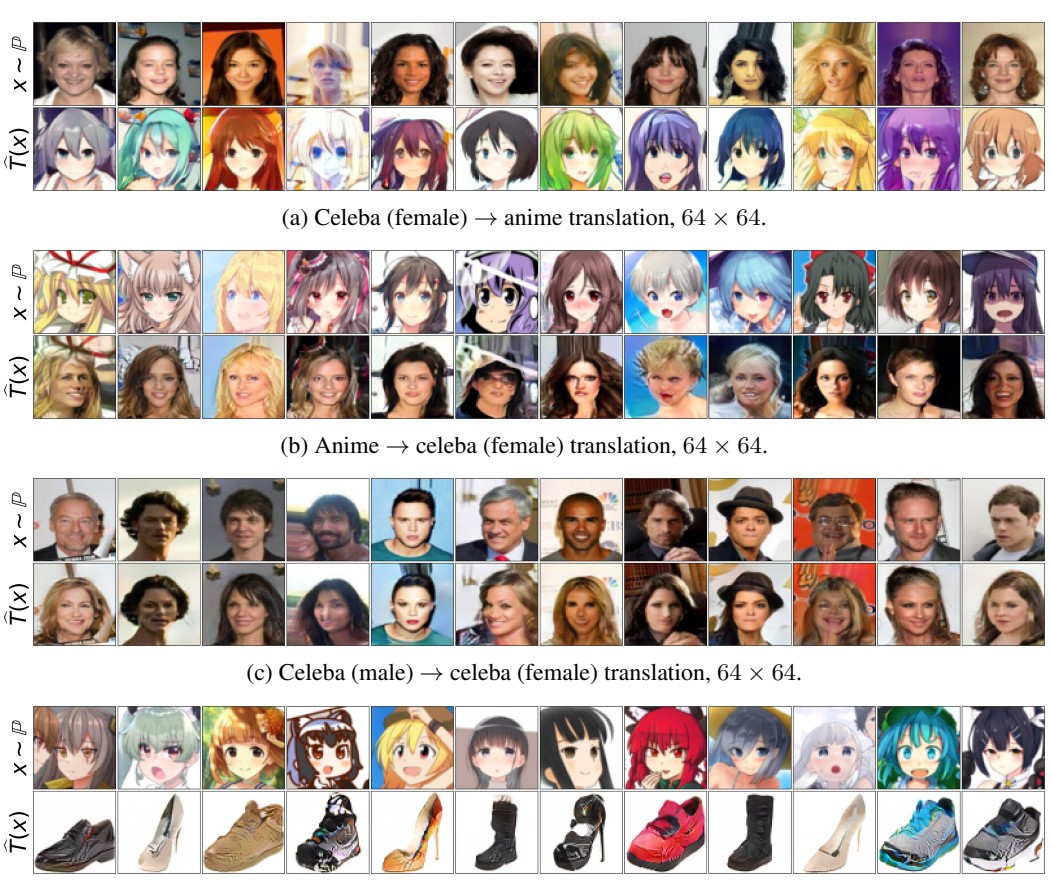

(a) Celeba (female) $\to$ anime translation, $64 \times 64$.

(b) Anime $\to$ celeba (female) translation, $64 \times 64$.

(c) Celeba (male) $\to$ celeba (female) translation, $64 \times 64$.

(d) Anime $\to$ shoes translation, $64 \times 64$.

Figure 16: Unpaired translation with OT maps ($\mathbb{W}_2$). Additional examples (part 2).

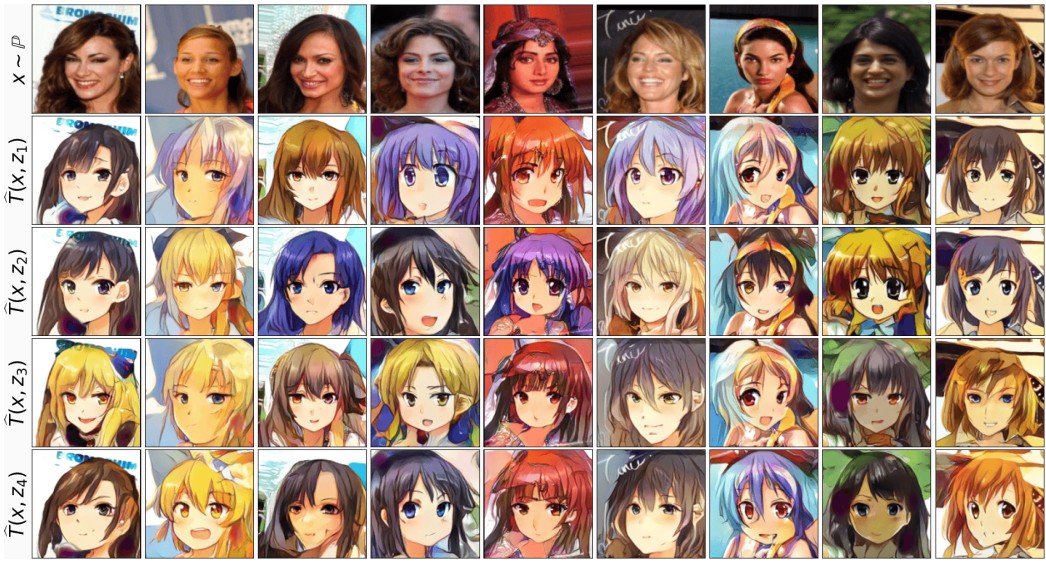

(a) Input images $x$ and random translated examples $T(x, z)$.

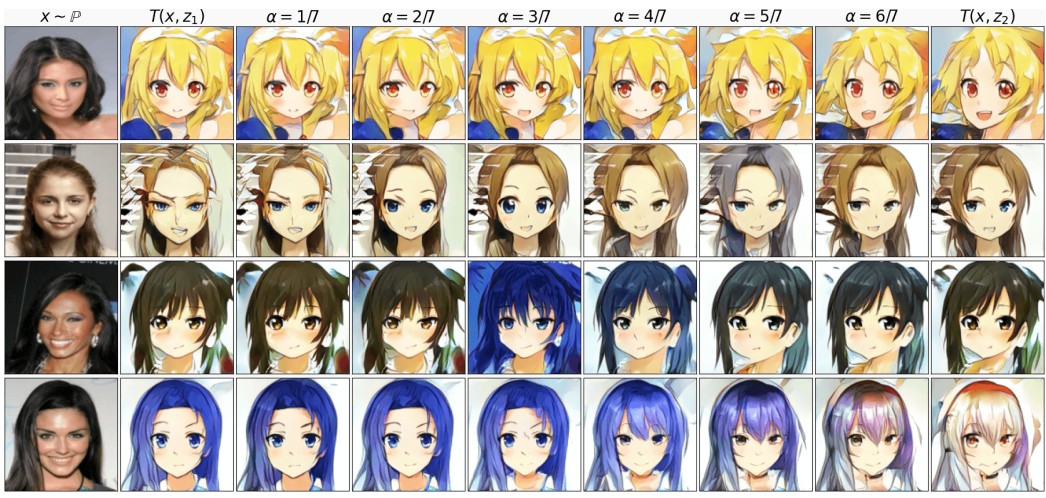

(b) Interpolation in the conditional latent space, $z = (1 - \alpha)z_1 + \alpha z_2$.

Figure 17: Celeba (female) $\rightarrow$ anime, $128 \times 128$, stochastic. Additional examples.

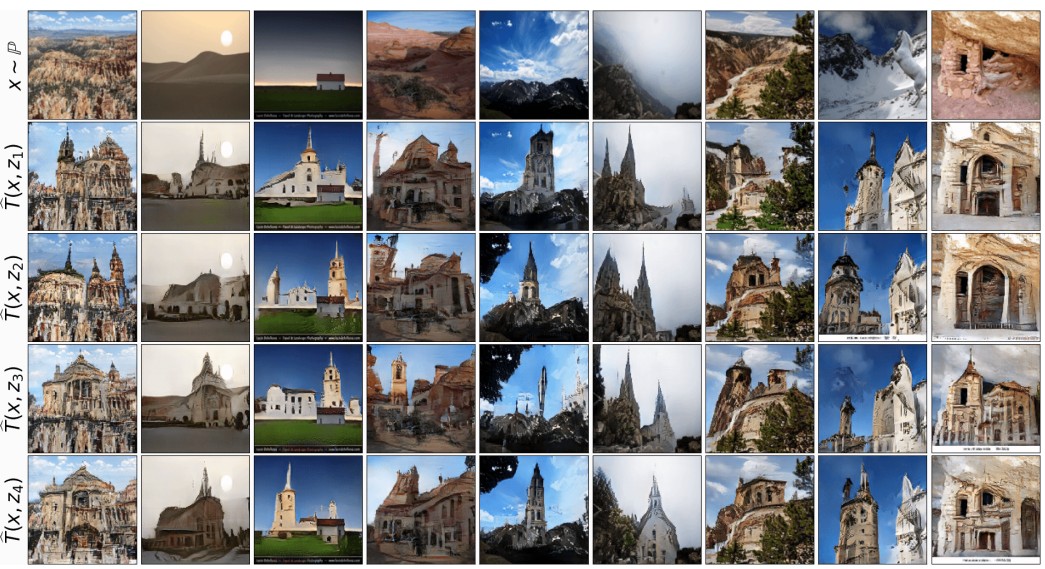

(a) Input images $x$ and random translated examples $T(x, z)$.

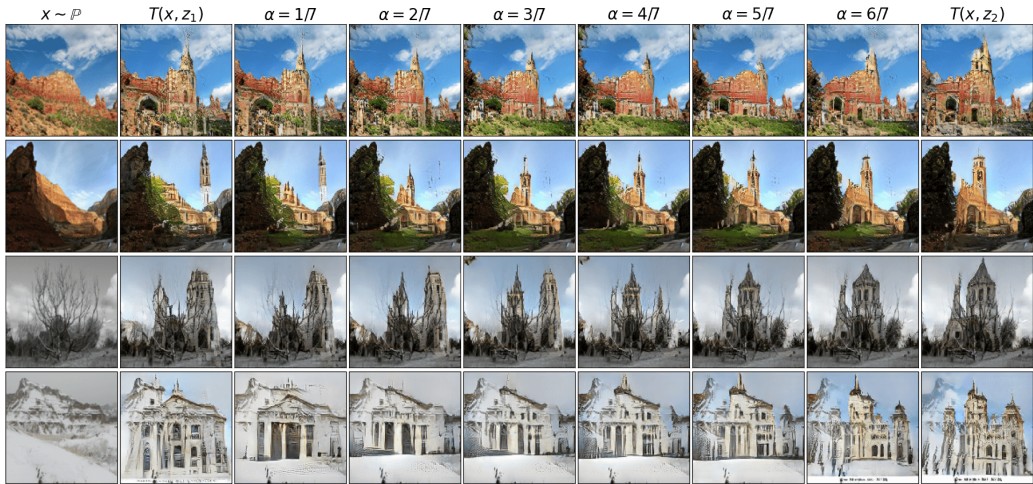

(b) Interpolation in the conditional latent space, $z = (1 - \alpha)z_1 + \alpha z_2$.

Figure 18: Outdoor $\rightarrow$ church, $128 \times 128$, stochastic. Additional examples.

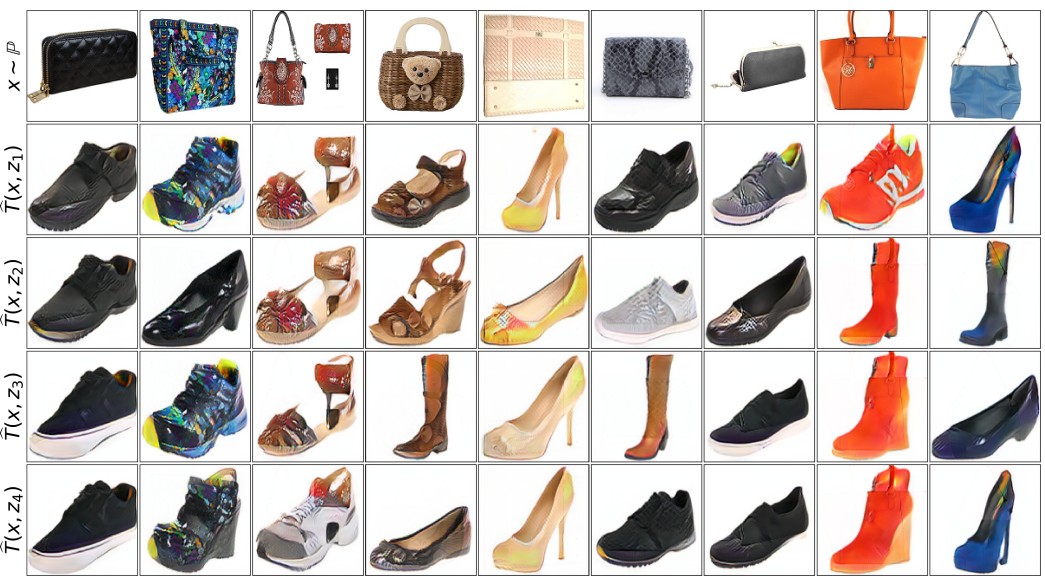

(a) Input images $x$ and random translated examples $T(x, z)$.

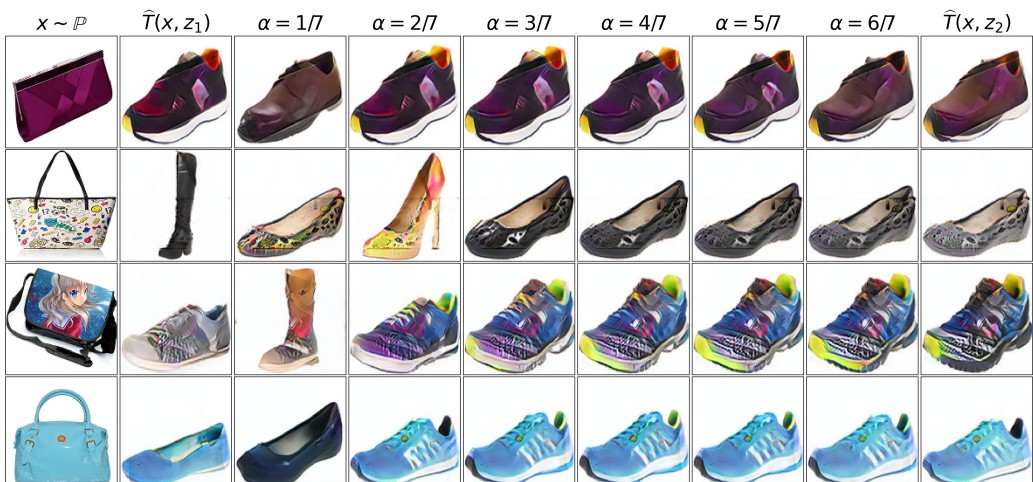

(b) Interpolation in the conditional latent space, $z = (1 - \alpha)z_1 + \alpha z_2$.

Figure 19: Handbags $\rightarrow$ shoes translation, $128 \times 128$, stochastic. Additional examples.

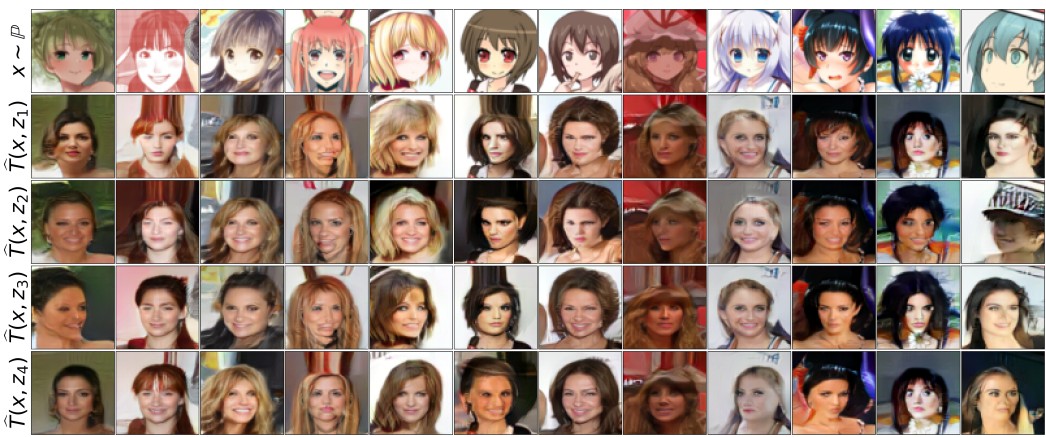

(a) Input images $x$ and random translated examples $T(x, z)$.

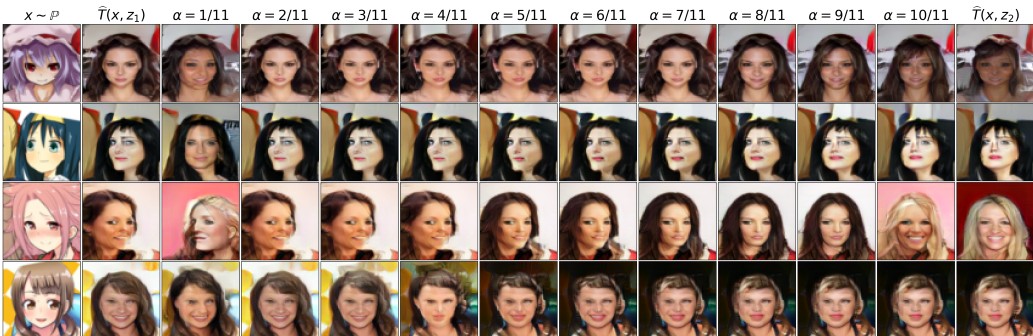

(b) Interpolation in the conditional latent space, $z = (1 - \alpha)z_1 + \alpha z_2$.

Figure 20: Anime $\rightarrow$ celeba (female) translation, $64 \times 64$, stochastic. Additional examples.

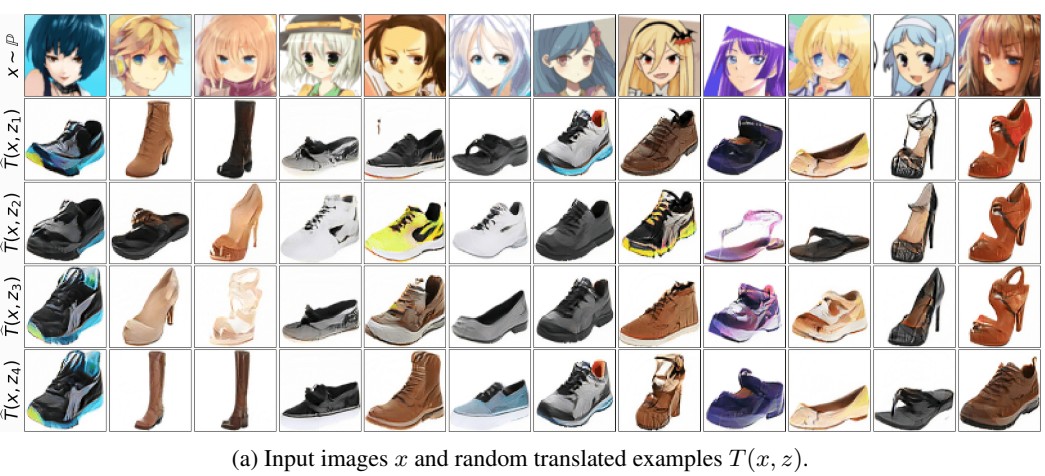

(a) Input images $x$ and random translated examples $T(x, z)$.

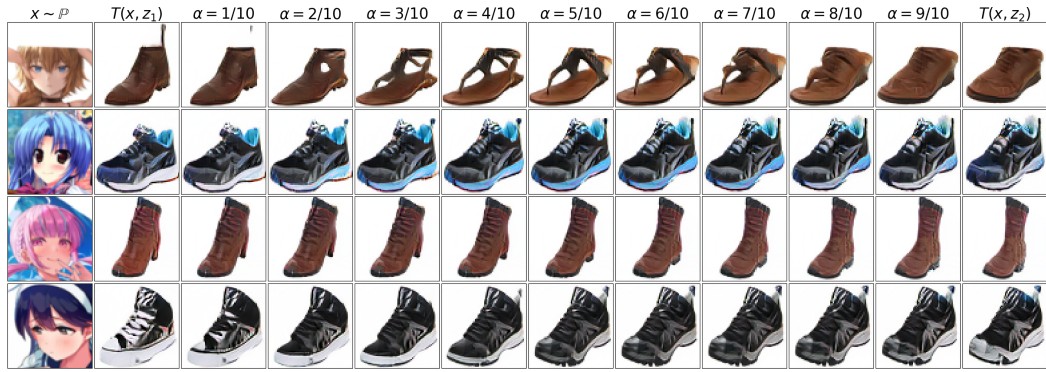

(b) Interpolation in the conditional latent space, $z = (1 - \alpha)z_1 + \alpha z_2$.

Figure 21: Anime $\rightarrow$ shoes translation, $64 \times 64$, stochastic. Additional examples.

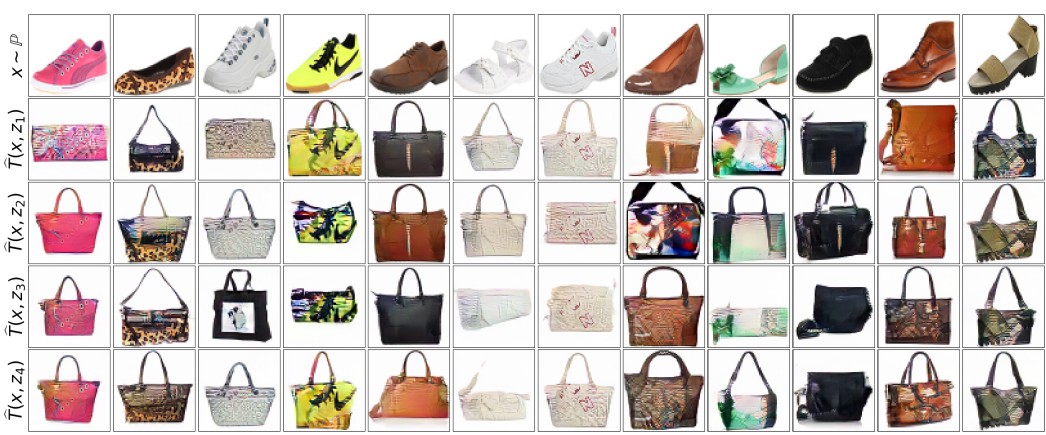

(a) Input images $x$ and random translated examples $T(x, z)$.

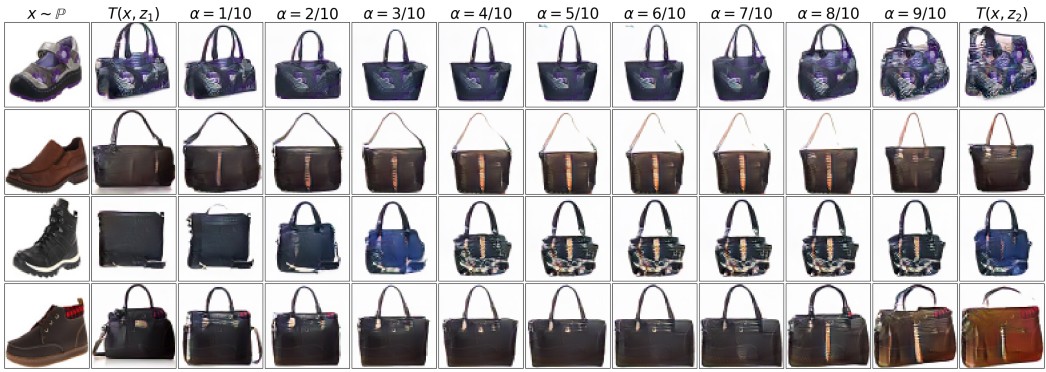

(b) Interpolation in the conditional latent space, $z = (1 - \alpha)z_1 + \alpha z_2$.

Figure 22: Shoes $\rightarrow$ handbags, $64 \times 64$, stochastic. Additional examples.

# I EXAMPLES WITH THE SYNCHRONIZED NOISE

In this section, for *handbags→shoes* (64×64) and *outdoor→church* (128×128) datasets, we pick a batch of input data $x_1, \ldots, x_N \sim \mathbb{P}$ and noise $z_1, \ldots, z_K \sim \mathbb{S}$ to plot the $N \times K$ matrix of generated images $T_\theta(x_n, z_k)$. Our goal is to assess whether using the same $z_k$ for different $x_n$ leads to some shared effects such as the same form a generated shoe or church.

The images results are given in Figures 23 and 24. Qualitatively, we do not find any close relation between images produced with the same noise vectors for different input images $x_n$.

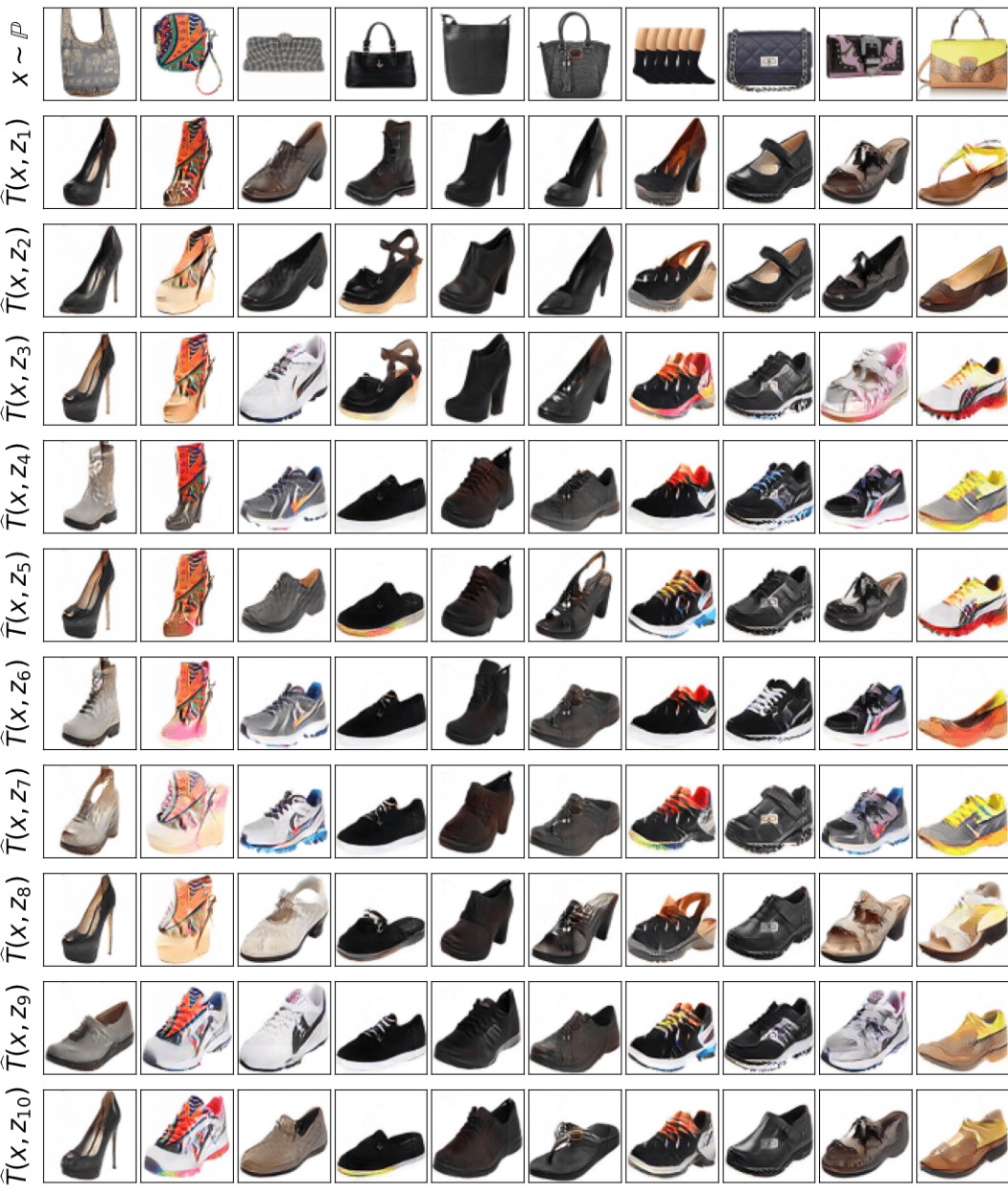

Figure 23: Input images $x$ and random translated examples $T(x, z)$.
In this example, noise inputs $z$ are synchronized between different input images $x$.

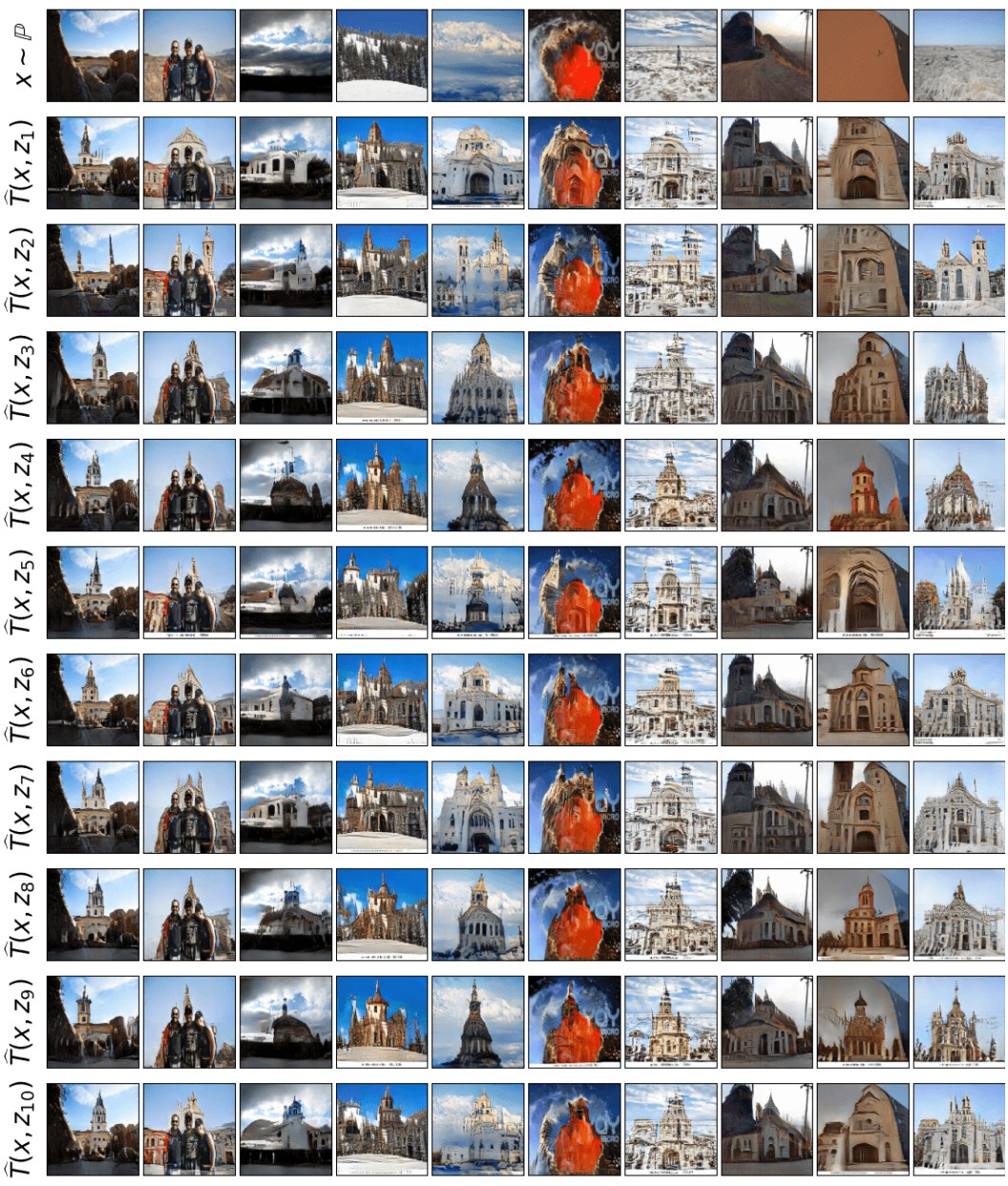

Figure 24: Input images $x$ and random translated examples $T(x, z)$.
In this example, noise inputs $z$ are synchronized between different input images $x$.

