# OpenReview forum: "Neural Optimal Transport"
_ICLR.cc/2023/Conference — ICLR 2023 notable top 25%_

### Official Review · Reviewer_NG9Z · 2022-10-20

**Confidence:** 4
**Correctness:** 3
**Technical Novelty And Significance:** 2
**Empirical Novelty And Significance:** 3
**Recommendation:** 6

**Clarity, Quality, Novelty And Reproducibility:**

Overall, I think this work is of high quality. Parameterizing the optimal transport estimators as neural networks following the Kantorovich duality is not entirely new but incorporating this weak-cost is quite novel.
Recently, diffusion models have been achieving incredible performances, especially in image-2-image transformation. The authors may want to justify the advantages brought by an optimal map compared with these approaches.

**Details Of Ethics Concerns:**

I have no concerns.

**Strength And Weaknesses:**

Pros:
- Overall, the paper is well-written as the authors delivered their proposed approach clearly.
- The organization of section 4 is encouraging and easy to follow. It is clear that the proposed method is motivated by the Kantorovich dual formulation with an additional C-transformation.
- The experimental part is quite sufficient. The authors have conducted experiments on multiple datasets, including anime faces, CelebA datasets and so on. This is quite valuable as OT map estimation approaches are usually concerned with scalability.

Cons and questions:
- Is it possible to better justify the advantages of the optimal transport map formulation, rather than existing deep generative frameworks? An optimal transport map may allow for better interpretability and generalization. Is it possible to have so called out-of-sample transport results, like in figure 1 of[1], figure 4 of [2]? Also, given the ideally nice smoothness properties of OT maps, shall we expect some results like "king - man + women - queen" when we are transporting image samples?
- While the problem formulation doesn't specify the ground metric. It seems that in the experiments, the authors are still using the quadratic cost. It is reasonable for the image datasets, however it could be nice if there are experimental results for other types of ground metrics, such as cosine similarity for language embeddings.
- What are the values of z_1, z_2, and z_3 in figure 6, 7, 12, 17 and later?  Does that impose some kind of interpretability on the map?
- From my understanding, it is true that under the given conditions, an optimal transport plan induced the transport map, which further characterizes the corresponding Wasserstein barycenters and the geodesic. However, given that the transport map is parameterized by a feedforward neural network (U-net), how does this U-net characterize the joint distribution of two distributions? In toy-1D experiments, the authors visualize the optimal plan but that is given by the ot.weak package.
In addition, the discriminator in a GAN can be viewed as a classifier. Are there any interpretations we can draw from the potential network f?

[1] Perrot, Michaël, et al. "Mapping estimation for discrete optimal transport." Advances in Neural Information Processing Systems 29 (2016).

[2] Zhu, Jiacheng, et al. "Functional optimal transport: map estimation and domain adaptation for functional data." arXiv preprint arXiv:2102.03895 (2021).


**Summary Of The Paper:**

This paper proposed a neural network based solver which estimates an explicit transport map rather than just using OT losses. Specifically, the authors introduced a weak OT formulation so as to find the potentially stochastic optimal transport map. The resulting objective becomes a minimax optimization which is learning a transport map and a potential simultaneously and both of them can be parameterized as neural networks. The proposed method is evaluated on a couple of datasets and compared against several baselines. The experimental results are quite encouraging as the estimated map can pushforwards image samples among different contexts but keep the style.

**Summary Of The Review:**

In this paper, the authors proposed a framework to estimate the optimal transport map parameterized as a neural network. The paper is well-written, and the contributions are clearly presented. The proposed weak OT cost extends the existing Kantorovich dual formulation, and yields a practical objective for training an OT map on large-scale datasets. The experimental result is quite encouraging.
Thus, I tend to recommend the acceptance of this work.

---

> ### Author Response · Authors · 2022-11-12
> **Answer to Reviewer NG9Z (part 1)**
>
> Thank you for spending time reviewing our paper and providing useful feedback that will help us improve the manuscript. Please find the answers to your questions below.
>
> **(1) Is it possible to better justify the advantages of the optimal transport map formulation, rather than existing deep generative frameworks?**
>
> Following your suggestion, we updated the **potential impact** paragraph in Section 6. We emphasized the advantage of better interpretability and, additionally, highlighted the fact that unlike most GANs or diffusion models, our method allows to easily control the required amount of diversity in generated samples (see Appendix A for examples).
>
> **(2) Is it possible to have so called out-of-sample transport results, like in figure 1 of [1], figure 4 of [2]?**
>
> We learn a network $T_{\theta}$ to represent the transport map. As a result, it can be applied to out-of-train-sample inputs $x$. As we write in Section 5 (*train-test-split*), the network $T_{\theta}$ is learned on the **train** samples, but all the qualitative and quantitative results for the image-to-image translation are given for the **test** samples which differ from the train ones.
>
> **(3) Also, given the ideally nice smoothness properties of OT maps, shall we expect some results like "king - man + women - queen" when we are transporting image samples?**
>
> To answer your question, we conducted the following experiment on *handbags*$\rightarrow$*shoes*  (64$\times$64) and *outdoor*$\rightarrow$*church* datasets (128$\times$128). We picked a batch of data $x_{1},\dots,x_{N}\sim\mathbb{P}$ and noise $z_{1},\dots,z_{K}\sim\mathbb{S}$ and plotted the $N\times K$ matrix of generated images $T_{\theta}(x_n,z_k)$ to see whether using the same $z_k$ for different $x_n$ leads to some shared effects such as the same form a generated shoe or church. **The images results are given in the newly added Appendix I.** Qualitatively, we do not see any close relation between images generated from the same latent vectors $z$ for different input images $x_{n}$.
>
> **(4) While the problem formulation doesn't specify the ground metric. It seems that in the experiments, the authors are still using the quadratic cost. It is reasonable for the image datasets, however it could be nice if there are experimental results for other types of ground metrics, such as cosine similarity for language embeddings.**
>
> We use the quadratic transport costs because they already provided a meaningful performance on the downstream task of unpaired image-to-image translation (the key testbed of our paper). We agree that tasks beyond image-to-image translation with the quadratic cost, e.g., the suggested cosine similarity-based alignment of language embeddings, are promising for our method. However, such applications require external components (language encoder/decoder) which not only further complicate the evaluation but, more importantly, might make it not transparent (the final performance will also depend on non-OT-related components). Therefore, we leave such applications for future studies.
>
> **(5) What are the values of $z_1, z_2, z_3$ in figure 6, 7, 12, 17 and later? Does that impose some kind of interpretability on the map?**
>
> The values $z_{1},z_{2},z_{3}$ are random (non-synchronized between different images $x$) realizations of the input noise. Samples $T_{\theta}(x,z)$ correspond to random samples from the learned conditional distribution $\pi^{*}(y|x)$ of an OT plan, see below.
>
> **(6) However, given that the transport map is parameterized by a feedforward neural network (U-net), how does this U-net characterize the joint distribution of two distributions?**
>
> The transport network $T_{\theta}$ implicitly represents the family of conditional distributions $\pi^{\star}(y|x)$ of an OT plan $\pi^{\star}(x,y)=\pi^{\star}(x) \pi^{\star}(y|x)$. This means that for a given $x\sim \pi^{\star}(x)$, one may obtain samples from $\pi^{\star}(y|x)$ by using $T_{\theta}(x,z)$ for random $z\sim\mathbb{S}$. Note that recovering the marginal distribution $\pi^{\star}(x)$ is not needed as by definition of the OT problem $d\pi^{\star}(x)\equiv d\mathbb{P}(x)$. The samples $x\sim\mathbb{P}$ are already available (it is just the input dataset).

---

> > ### Author Response · Authors · 2022-11-12
> > **Answer to Reviewer NG9Z (part 2)**
> >
> > **(7) In addition, the discriminator in a GAN can be viewed as a classifier. Are there any interpretations we can draw from the potential network f?**
> >
> > Optimal transport tasks (2) and (3) are optimization problems over probability distributions $\pi(x,y)$ on $\mathcal{X}\times\mathcal{Y}$ with two boundary constraints: $d\pi(x)\equiv d\mathbb{P}(x)$ and $d\pi(y)\equiv d\mathbb{Q}(y)$. The dual potential $f$ can be viewed as the Lagrange multiplier for the **second** boundary constraint $d\pi(y)\equiv d\mathbb{Q}(y)$. This kind of interpretation mostly appears in discrete OT, e.g., see Proposition 24 in  [this guide](https://mathematical-tours.github.io/book-sources/optimal-transport/CourseOT.pdf), where the Lagrange multipliers for **both** the boundary constraints are considered for the entropy-regularized OT problem. In the discrete case, the multipliers are just vectors (not functions) as the supports of distributions are finite.
> >
> > Being the Lagrange multiplier, the dual potential $f$ is a meaningful function containing information about the OT problem. For example, in the case of strong OT with the quadratic cost, one may recover the **inverse** OT map ($\mathbb{Q}\rightarrow\mathbb{P}$) from the dual potential $f$ as $x-\nabla f^{*}(x)$, see discussion in Section 1 in (Korotin et al., 2021b). Analogous statement works for more general strong costs as well, see Theorem 1.17 in (Santambrogio, 2015). We do not know whether such observations hold true for weak costs.
> >
> > **Concluding remarks**. Please respond to our post to let us know if the clarifications above suitably address your concerns about our work. We are happy to address any remaining points during the discussion phase; if the responses above are sufficient, we kindly ask that you consider raising your score.

---

### Official Review · Reviewer_D9LF · 2022-10-23

**Confidence:** 3
**Correctness:** 4
**Technical Novelty And Significance:** 3
**Empirical Novelty And Significance:** 3
**Recommendation:** 8

**Clarity, Quality, Novelty And Reproducibility:**

The work is very well written, the key concepts, mathematical formulations, are highly motivated and clearly represented; the deduction process is summarized as a sequence of lemmas, which are easy to follow and very inspiring; the explanation of the key idea of noise out sourcing is very elegant and exciting; the experimental results are thorough and convincing.

The method is relatively novel. Although there are so many works addressing on the efficient computation of OT maps/plans, the one using noise outsourcing is quite novel.

The quality of the work is good, the mathematical modeling and formulation, the usage of theoretical tools are rigorous, and numerical experiments are concrete.

The authors promise to make their code public available, hence the work should be reproducible.

**Details Of Ethics Concerns:**

The work is a fundamental research, it focuses on theoretic exploration.

**Strength And Weaknesses:**

The work has the following strengths:

1. The problem to tackle is to efficiently compute optimal transport maps or plans, which play fundamental roles in deep learning and are highly non-linear and notoriously hard to compute.
2. By using the noise outsourcing idea, the problem is converted to a minmax optimization problem, which can be tackled efficiently
3. The experimental results demonstrate the effectiveness of the proposed modal

The weakness is that the proposed algorithm searches for a solution of a saddle point problem and extract the stochastic OT map from it, but not all saddle points correspond to optimal stochastic OT maps, this type of ambiguity requires further theoretical exploration.

**Summary Of The Paper:**

This work presents a novel neural-networks-based algorithm to compute optimal transport maps and plans for both strong and weak transport costs. The problem is converted to a minmax optimization one using noise outsourcing idea. The work proves that the proposed neural-networks are universal approximators of transport plans between probability distributions. The algorithm is evaluated on both toy examples and on unpaired image-to-image translation.

The contributions of the work are
1. proposes a novel framework to compute optimal transport maps and plans for both strong and weak transport costs based on noise outsourcing method;
2. give rigorous proofs for the existence of the solution and show the proposed model is a universal approximator
3. thorough experimental results to demonstrate the effectiveness of the proposed model

**Summary Of The Review:**

This work presents a novel neural-networks-based algorithm to compute optimal transport maps and plans for both strong and weak transport costs. The problem is converted to a minmax optimization one using noise outsourcing idea. The work proves that the proposed neural-networks are universal approximators of transport plans between probability distributions. The algorithm is evaluated on both toy examples and on unpaired image-to-image translation.

It is well known that optimal transport maps/plans play fundamental roles in deep learning, but their computations are intrinsically complicated. An efficient and accurate computational method is highly desirable. This work offers a novel method for tackling this fundamental problem by converting it as a min-max optimization problem using the noise outsourcing idea. The algorithm can compute both deterministic and stochastic mappings with strong and weak transport costs. The various applications show the flexibility and effectiveness of the method. Although there are some theoretic ambiguity for the saddle points, the proposed method is inspiring the convincing.

---

> ### Author Response · Authors · 2022-11-12
> **Answer to Reviewer D9LF**
>
> Dear reviewer, thanks for your positive review! We are inspired by the fact that your highly appreciate our contribution and find our method inspiring and convincing.
>
> We agree with your comment that the theoretical ambiguity in saddle points might require more exploration. To form the basis for the future exploration, we provide a sufficient condition on the weak cost $C(x,\mu)$ to completely remove this ambiguity, see Lemma 5 in Appendix F. Developing ways to satisfy this condition is a promising future research direction.

---

### Official Review · Reviewer_KZKB · 2022-10-24

**Confidence:** 3
**Correctness:** 4
**Technical Novelty And Significance:** 3
**Empirical Novelty And Significance:** 3
**Recommendation:** 8

**Clarity, Quality, Novelty And Reproducibility:**

This paper is of high clarity, quality and reproducibility.



**Strength And Weaknesses:**

The paper is very well-written and pedagogical. It also mixes a good deal of theoretical and algorithmic approaches along with thorough empirical evaluations. Maybe the only weakness is that it is mostly motivated by the situation where no deterministic map approaches, and while it is clear that it might be theoretically the case, it is less clear when that situation occurs in practice.


**Summary Of The Paper:**

This paper provides a scalable way to learn OT maps via a deep neural network. It is especially relevant due to the recent approaches demonstrating the use of OT maps for generative purposes, as opposed to the previous methods where OT was used as a loss when training generators.


**Summary Of The Review:**

This is a very good paper.

---

> ### Author Response · Authors · 2022-11-12
> **Answer to Reviewer KZKB**
>
> Dear reviewer, thanks for your positive review! We are inspired by the fact that your highly appreciate our contribution.
>
> Regarding your comment about the non-existence of the deterministic map. Such situations are common in image restoration problems, e.g., image super-resolution, which are mostly ill-posed: for the input degraded image there usually exists several possible restorations, see [1, 2]. Although we do not consider such applications in the paper, we think that our generic method can be adapted to such setups as well.
>
> **References**
>
> [1] Lugmayr, A., Danelljan, M., \& Timofte, R. (2021). NTIRE 2021 learning the super-resolution space challenge. In Proceedings of the IEEE/CVF Conference on Computer Vision and Pattern Recognition (pp. 596-612).
>
> [2] Lugmayr, A., Danelljan, M., Gool, L. V., \& Timofte, R. (2020, August). Srflow: Learning the super-resolution space with normalizing flow. In European conference on computer vision (pp. 715-732). Springer, Cham.

---

### Author Response · Authors · 2022-11-12
**Revised Paper**

Dear reviewers, thanks for your insightful comments! We highly appreciate that you all are positive about our work. We have uploaded an updated version of the paper. The changes are highlighted with the **blue color**. The main changes are:

**[NG9Z]** In Section 6, we emphasized the advantages (better interpretability, controllable diversity) of our method w.r.t. popular models based on GANs or diffusion models.

**[NG9Z]** In Section 5.3, we added a remark that the random vectors $z_{1}, z_{2}, z_{3}$ are not synchronized, i.e., they differ for different inputs $x$. We added **Appendix I** with the example of images $T(x,z)$ generated for synchronized $z$.

Please consider the updated revision. We are happy to address any remaining points during the discussion phase.

---

### Decision · Program_Chairs · 2023-01-20

**Decision:**

Accept: notable-top-25%

**Justification For Why Not Higher Score:**

min/max formulation and stochastic parametrization for OT problems  have been proposed in other works previously in the unbalanced setting this paper complements previous work in the strong and weak OT setup  and the optimal transport community will benefit more from hearing about this work for applications of OT in the continuous setting.

**Justification For Why Not Lower Score:**

The paper is of high quality and all reviewers agreed on its acceptance and its quality.

**Metareview: Summary, Strengths And Weaknesses:**

The paper proposes a min-max formulation to solve strong a weak transport cost with a stochastic parametrization of the transport plan. There are some similarity of the proposed method and adversarial learning of unbalanced transport maps https://arxiv.org/abs/1810.11447 . Authors clarified the differences and should discuss this work in their paper.

The paper evaluates the proposed methods on synthetic and unpaired image to image style translation tasks.

**Note From Pc:**

if the above contains the word "oral" or "spotlight" please see: "oral" presentation means -> notable-top-5% and "spotlight" means -> notable-top-25%. As stated in our emails, we are disassociating presentation type from AC recommendations

**Summary Of Ac-Reviewer Meeting:**

N/A